# Neighbor-enhanced diffusivity in dense, cohesive cell populations

**Hyun Gyu Lee** **, Kyoung J. Lee** *

Department of Physics, Korea University, Seoul, Korea

* kyoung@korea.ac.kr

**Data Availability Statement:** All relevant data for the cell trajectories of experiments, simulations, and the codes for the analysis are available on https://drive.google.com/drive/folders/1ce0EYLr3yZneZWVl03rejUrISF2tTdgL?usp=

## Abstract

The dispersal or mixing of cells within cellular tissue is a crucial property for diverse biological processes, ranging from morphogenesis, immune action, to tumor metastasis. With the phenomenon of 'contact inhibition of locomotion,' it is puzzling how cells achieve such processes within a densely packed cohesive population. Here we demonstrate that a proper degree of cell-cell adhesiveness can, intriguingly, enhance the super-diffusive nature of individual cells. We systematically characterize the migration trajectories of crawling MDA-MB-231 cell lines, while they are in several different clustering modes, including freely crawling singles, cohesive doublets of two cells, quadruplets, and confluent population on two-dimensional substrate. Following data analysis and computer simulation of a simple cellular Potts model, which faithfully recapitulated all key experimental observations such as enhanced diffusivity as well as periodic rotation of cell-doublets and cell-quadruplets with mixing events, we found that proper combination of active self-propelling force and cell-cell adhesion is sufficient for generating the observed phenomena. Additionally, we found that tuning parameters for these two factors covers a variety of different collective dynamic states.

## Author summary

Dispersal or movement of cells within dense biological tissue is essential for diverse biological processes, ranging from pattern formation, immune action, to tumor metastasis. However, it is quite puzzling how cells acquire such ability when they are supposedly "caged" by neighboring cells. Here, we report an unusual property of (MDA-MB-231) breast cancer cells that diffuse more persistently within a densely packed population than when they are free to crawl around with little interference. This property is rather surprising since they prefer to stick together, forming clusters. Interestingly, however, we find that having sticky neighbors not only makes two active cells in contact periodically rotate, reminiscent of a ballroom dance, but also enhances the persistence of the cells within a dense population. These intriguing phenomena appear to be universal as they can be generated by a simple cellular Potts model with appropriate combination of active self-propulsion and cell-cell adhesion force.

sharing Or https://github.com/josephlee188/Data-for-PLOS-Comp.-Biol.-publication.

**Funding:** This work was supported by the National Research Foundation of Korea (NRF) grant funded to KJL by the Korea government(MEST) (No. 2019R1A2C2005989). The funders had no role in study design, data collection and analysis, decision to publish, or preparation of the manuscript.

**Competing interests:** The authors have declared that no competing interests exist.

# Introduction

Biological cells are the fundamental building blocks of all lifeforms, ranging from single-cell animal-like amoeba to more complex multi-cellular organisms like us human beings, in which they form various organs and complex physical structures. Indeed, cells are a remarkable material with an amazingly wide range of versatility and flexibility. Some densely packed populations of cells are rigid enough to be viewed as a solid (for example, imagine a piece of hardwood or animal bone), while in some other cases cells prefer to stay alone and move erratically, like gas particles as in populations of Dictyostelium discoidium amobae in their vegetative state [1] or microglia (immune cells of the brain) which tend to avoid each other upon a physical contact in culture [2]. Cells within a solid-like population are caged by their immediate neighbors, maintaining their relative positions and orientations [3–8]. By far more common situation is that cells form soft tissues or tissue-like structures which are analogous to amorphous materials like glass, or perhaps, more accurately, something in between glass and liquid in which cells can diffuse, flow, and flock [3–10].

Ice melts to water, and water evaporates to vapor, as the temperature rises: In other words, the state of matter transforms as some system parameter(s) change. Likewise, the biophysical state of the cellular population can also undergo transitions [11], and the topic has drawn a great deal of attention in biology in association with morphogenesis and tissue-remodeling [12–14], wound repairing [15,16], and tumor growth [17,18]. Biological tissues are a collection of interacting 'active' cells in nonequilibrium states, therefore, in principle, cell-population can support numerous different non-epithelial dynamic states, to which an initially sedentary tissue state can switch. One of such states is flocking and a good example is the recent experimental study of Mitchel et al. [19], where air pressure was used as a control parameter for transforming an airway epithelial tissue to cooperatively migratory flocking cells. The authors referred to the transformation as an unjamming transition (UJT). The same study could also induce a different type of transition, which the authors termed as partial epithelial-to-mesenchymal transition (pEMT): Here, a chemical agent called TGF-β1 was used to transform an initial epithelial state into a hybrid state having a mixture of epithelial and mesenchymal characteristics. The authors have concluded that UJT and pEMT, respectively, yielded two rather different liquid-like states having very divergent dynamic, structural and molecular marker characteristics. The study of Mitchel *et al.* is an excellent example suggesting that for fully unfolding the dynamics of the dense cell population the relevant phase space needs to be at least two-dimensional (i.e., requires two independent parameters).

Until now, a great deal of attention has been paid mainly to the (phase space) parameter regime in which interesting large-scale collective waves and flocks could form [16,20–22]. In the meanwhile, a more prevalent condition, especially for tumors, could involve the pEMT states near the border to sedentary epithelial states. Unlike a large-scale flock, whose spatial correlation length spans multiple cell-diameters, pEMT states may have a very small range of cell-cell interaction [3,23] with a spatial correlation length just about the size of a single-cell. So far, the spatiotemporal population dynamics within this regime is not well characterized at all except for the fact that cellular rearrangement can occur via cage-breaking with junctional change and rearrangements [5,24].

In this paper, we showed that cells in a densely packed two-dimensional cell layer in a hybrid (pEMT) regime could support some remarkable motile properties: They exhibited a higher diffusivity and longer persistent time in the moving direction than those of freely crawling single cells. Moreover, we demonstrated that the fascinating phenomena observed in small cell-clusters, namely 'cell-doublet rotation' and 'position swapping in cell-quadruplet', account for this dynamic property. The set of unusual observations was initially made with two-

dimensional cell cultures of MDA-MB-231 breast cancer cell lines. Then, they were reproduced faithfully with a cellular Potts Model (CPM), in which we tuned two key parameters associated with cell-to-cell adhesiveness (or stickiness) and self-propulsion, in a systematic fashion. Importantly, we identified the relevant parameter regime, where the CPM simulations reproduced the core experimental observations in a self-consistent way. Interestingly, a significant amount of cell-cell adhesion was mandatory for the observed enhancement of persistence and diffusivity of cells. The phenomenon of neighbor-enhanced diffusivity (NED) of confluent population of MDA-MB-231 has a strong similarity to 293-MOCA expressing human kidney 293T cells, which become super-diffusive only in the presence of cell-cell adhesive interactions [25].

## Results

### MDA-MB-231 cell-population vs. freely crawling cells

MDA-MB-231 cell-lines are a famous triple-negative breast cancer cell known for their aggressive, metastatic potential [26–28]. We chose MDA-MB-231 cell lines for our investigation since they have been popularly used for many different cancer studies. More importantly, recent studies [29,30] clearly suggested that they pertain to the aforementioned hybrid epithelial-mesenchymal properties. Monoclonal MDA-MB-231 cells were cultured following a typical laboratory culture protocol, harvested, and uniformly plated on a culture dish for imaging (see Methods for more details). A small central square area of the densely packed population is shown as a black and white image on the x-y plane of Fig 1A. The same figure also illustrates exemplary trajectories of five moving cells that were initially located adjacent to each other in the middle of the population. As they diffused out, they intermittently made sharp turns and fluctuations in their instantaneous speeds (0 ~ 7 $\mu m$/min), which are represented by the thickness of the colored 'tubes' in Fig 1A. Similarly, Fig 1B illustrates a superposition of trajectories of five randomly chosen, *freely* moving cells that experienced little or no cell-cell interaction. In both cases, the cells were super-diffusive, with the diffusivity of the cells in densely packed population being even stronger than that of the freely moving cells. This property is apparent in Fig 1C and 1D. Fig 1C shows a log-log plot of mean squared displacement (MSD) $\langle \delta^2 \rangle$ versus time $t$, where solid red and blue dots trace the average profiles for freely crawling cells and cells in confluent population, respectively, while the background shadows represent the associated standard deviations. The upper panel of Fig 1D plots the diffusion exponent $\alpha$ for the marked time window in Fig 1C, which is blown up in lower Fig 1D. If we assume $\langle \delta^2 \rangle = Dt^\alpha$ with $D$ diffusion coefficient, $\alpha = 1$ stands for a normal diffusion, $\alpha < 1$ ($\alpha > 1$) represents a sub-diffusion (super-diffusion), and $\alpha = 2$ stands for a ballistically moving object. As the plot of $\alpha$ in Fig 1D clearly illustrates, the value of $\alpha$ for the cells in confluent population was significantly larger than that of freely crawling cells for the entire range of time. In addition, the time range of super-diffusivity ($\alpha > 1$) for freely migrating cells terminated around $t \sim 10^{2.4}$ (~ 251) min, quite earlier than the time, approximately $10^{2.7}$ (~ 501) min, beyond which cells in confluent population lose their super-diffusivity.

The difference in the degree of diffusivity discussed in Fig 1C and 1D was also consistent with the analysis of auto-correlation function of unit tangent vector, $c_{auto} = \langle \hat{e}(t_o) \cdot \hat{e}(t_o + t) \rangle$ (see Fig 1E), which measures the average degree of moving directional correlation between two unit-tangent vectors obtained at two different time instances which are separated by time $t$. The function $c_{auto}$ decayed significantly faster for freely migrating cells than for the cells in confluent populations (average decaying time constants of the ensemble: $\langle \tau_{2,free} \rangle = 66 \pm 104$ min vs. $\langle \tau_{2,pop} \rangle = 116 \pm 150$ min. In other words, cells in a confluent population had a stronger tendency of moving along (or a longer memory of) the direction to which they were heading

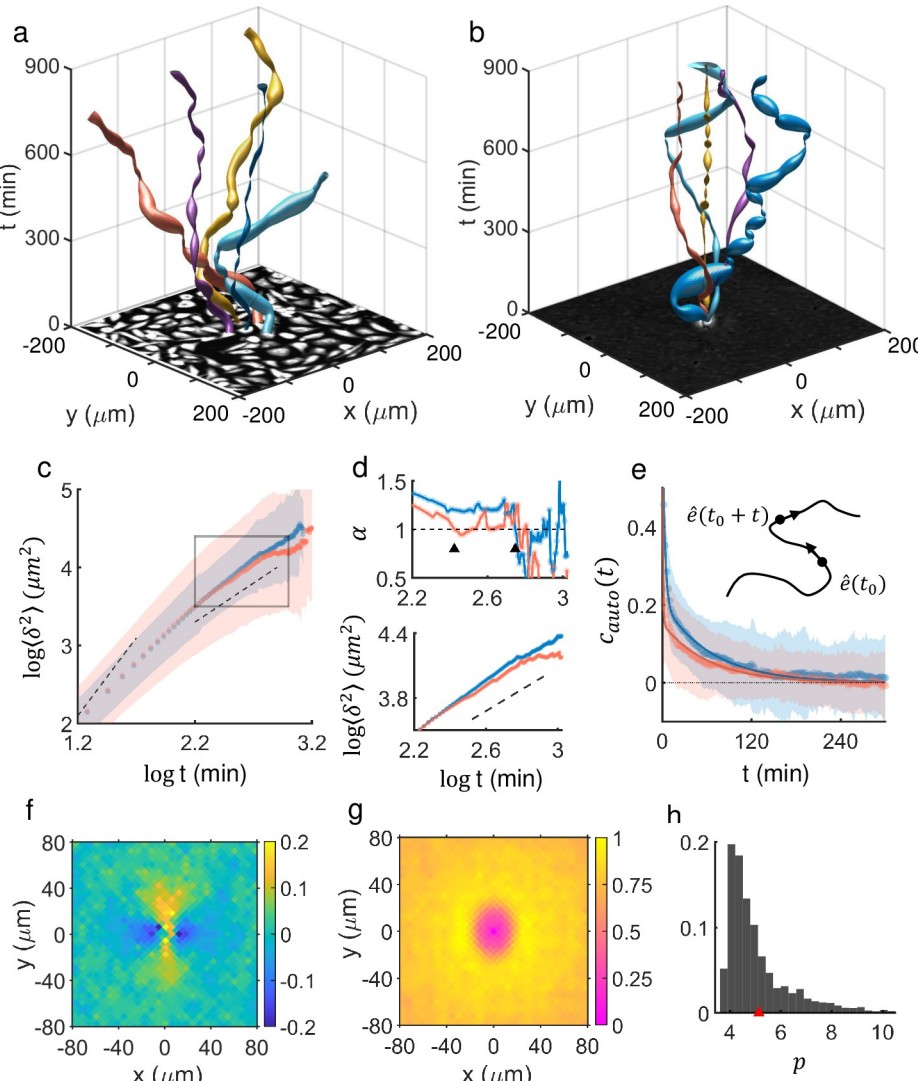

**Fig 1. Enhancement of diffusivity of MDA-MB-231 cells on a two-dimensional substrate in the presence of neighboring cells.** (a) Confluent population of MDA-MB-231 cells shown as a black/white image on the x-y plane at $t = 0$ and exemplary moving trajectories of 5 diffusing cells (S1 Movie). The thickness of each colored tube represents the cell's instantaneous moving speed. (b) Superposition of 5 representative trajectories of freely migrating cells that were prepared separately [all are positioned at the origin (0,0) at $t = 0$] (S2 Movie). (c) Log-log plot of MSD of cells in confluency (blue) and freely migrating cells (red) vs. time (n = 258 for confluent population and n = 187 for free, single cells. Blue/red dots are mean values, while the size of matching colored shadow represents standard deviation; and dashed lines mark slope of 2 and 1, respectively). Data points shown here are separated by 10 mins. (d) A plot of diffusion exponent $\alpha$ (top) and its matching plot of MSD (bottom, inset of (c)): Two dashed lines represent $\alpha = 1$, where $\alpha$ was computed by calculating the slope of MSD between 10 successive data points for smoothing and sliding across the time. For the time domain of interest ($log158 = 2.2 \sim log501 = 2.7$, the average $\overline{\alpha}_{free} = 1.07 < \overline{\alpha}_{confluent} = 1.24$ and the difference is statistically very significant (one-way ANOVA test p < 0.001). (e) Auto-correlation functions of unit tangent vectors of cells in confluency (blue), and freely migrating cells (red). The overlaid solid lines are a fit of two-tier exponential function $C(t) = Ae^{-t/\tau_1} + Be^{-t/\tau_2}$: for the cells in confluency, $(\tau_1, \tau_2) = (3.02\ min, 66.11\ min)$ and $(A, B) = (0.56, 0.21)$; and for freely migrating cells, $(\tau_1, \tau_2) = (0.81\ min, 62.10\ min)$ and $(A, B) = (0.84, 0.15)$. R-squared values are ~ 0.99 for both cases. The MSDs and the auto-correlation functions were calculated using data points separated by 2 min. (f) (Temporal and ensemble) average spatial correlation map of $c_{cross}$. A negative value of $c_{cross}$ means that the cell at the corresponding location at a given time has a preference of moving in the opposite direction to the reference cell located at (0,0). (g) Normalized (temporal and ensemble) average density map of neighboring cells in a confluent population. The oval shape (aspect ratio of 0.6) represents the average shape of a cell moving in positive y-axis. (h) Shape index histogram of the cells in confluency reflecting both temporal as well as ensemble inhomogeneities (mean value is 5.1 ± 1.4).

than freely migrating cells. Subsequently, considering the mean migration speed of $\langle v_{free} \rangle$ = 0.8 ± 0.3 $\mu m$/min and $\langle v_{pop} \rangle$ = 0.6 ± 0.2 $\mu m$/min, the matching persistence lengths are, respectively, $\langle l_{free} \rangle$ = 52 ± 81 $\mu m$ and $\langle l_{pop} \rangle$ = 72 ± 99 $\mu m$. According to [31], one may require $\alpha > 1$ for the time greater than the directional persistence time $\tau_2$ as a necessary condition for super-diffusivity. Since the transition time points to the normal diffusion were measured to be around 251 min (for freely crawling cells, indicated by the 1st arrow in Fig 1D upper panel) and around 501 min (for cells in confluency, indicated by the 2nd arrow), which are much larger than $\langle \tau_{2,free} \rangle$ = 66 min and $\langle \tau_{2,pop} \rangle$ = 116 min, respectively, it seems appropriate to use the term super-diffusivity.

The analyses given in Fig 1A–1D univocally pointed to the fact that the enhanced diffusivity was not at all due to a large-scale cooperative flocking movement. As a matter of fact, the range of the cell-to-cell interaction turned out to be rather short, spanning only about one cell-diameter ($\sim$ 30 $\mu m$). For this estimation, we identified all individual cells and their centroids for a stack of a time-lapse movie (700 frames, 2 mins interval) and computed mean spatial velocity-velocity correlation map of $c_{cross} = \langle \hat{e}(\overrightarrow{r}_o) \cdot \hat{e}(\overrightarrow{r}_o + \overrightarrow{r}) \rangle$ as shown in Fig 1F, where $\overrightarrow{r}_o$ and $\overrightarrow{r}$ represent the position of the reference cell and those of other cells in its vicinity, respectively. To generate Fig 1F, the position of each reference cell was shifted to the origin (0,0) and its instantaneous moving direction was rotated to align toward the positive y-axis [i.e., $\hat{e}(r_o \to 0) \parallel \hat{y}$]. Fig 1F also shows that, on average, immediate neighbors on both sides of a reference cell along the x-axis have a strong negative correlation indicating that the neighbors have a tendency of moving in the opposite direction of the reference cell is heading. The correlation lengths extracted by fitting the slices of Fig 1F along vertical (moving direction) and horizontal (perpendicular to the moving direction) lines (passing through the origin) with the exponential functions as done in [4] were 30.75 $\mu m$ and 18.08 $\mu m$ (S1 Fig), respectively, indicating that they amount to only a single-cell diameter. Given the velocity-velocity correlation map of Fig 1F, we may interpret that the cell motility in confluency is a random mixture of a few cells co-moving in line like a "two-cart vehicle" or rotating (i.e., moving past each other) together in a time-shared manner. Finally, Fig 1G depicts the overall shape of MDA-MB-231 cell in a confluent population. Simply, it is the average point density map of the centroids of all cells in a given population with respect to the reference cell, whose position was again brought to the origin (0,0) and whose instantaneous moving direction was aligned toward the positive y-axis as it was done for Fig 1F: The oval shape with its long axis along the y-axis is rather clear; this is because the migrating cells tended to elongate along the direction of movement. This oval shape hides the heterogeneities and time-fluctuations of individual cells' shapes within the population. In fact, despite the monoclonal nature, the shapes of MDA-MB-231 cell-lines in confluent culture were quite heterogeneous as clearly shown by the broad distribution function of shape index $p$ (defined as $perimeter/\sqrt{area}$) in Fig 1H (edges of the cells in confluency were obtained by thresholding the phase-contrast images and identifying contours in the binary images).

## Description of cell-doublets

A useful insight into the nature of NED within dense population could be obtained by analyzing the motile behavior of dispersed 'doublets.' Low-density cultures of MDA-MB-231 cells generally exhibited many dispersed pairs, as the cells divided following a cell-cycle (period $\sim$ 32 hr) and preferred to stay together during migration (see S2 Fig for details). Doublets could also be formed by random encounters of individual cells. Accordingly, we tracked multiple doublets and analyzed the moving trajectories of the cells involved. Remarkably, the MDA-MB-231 doublets in our culture almost invariably exhibited a robust rotational

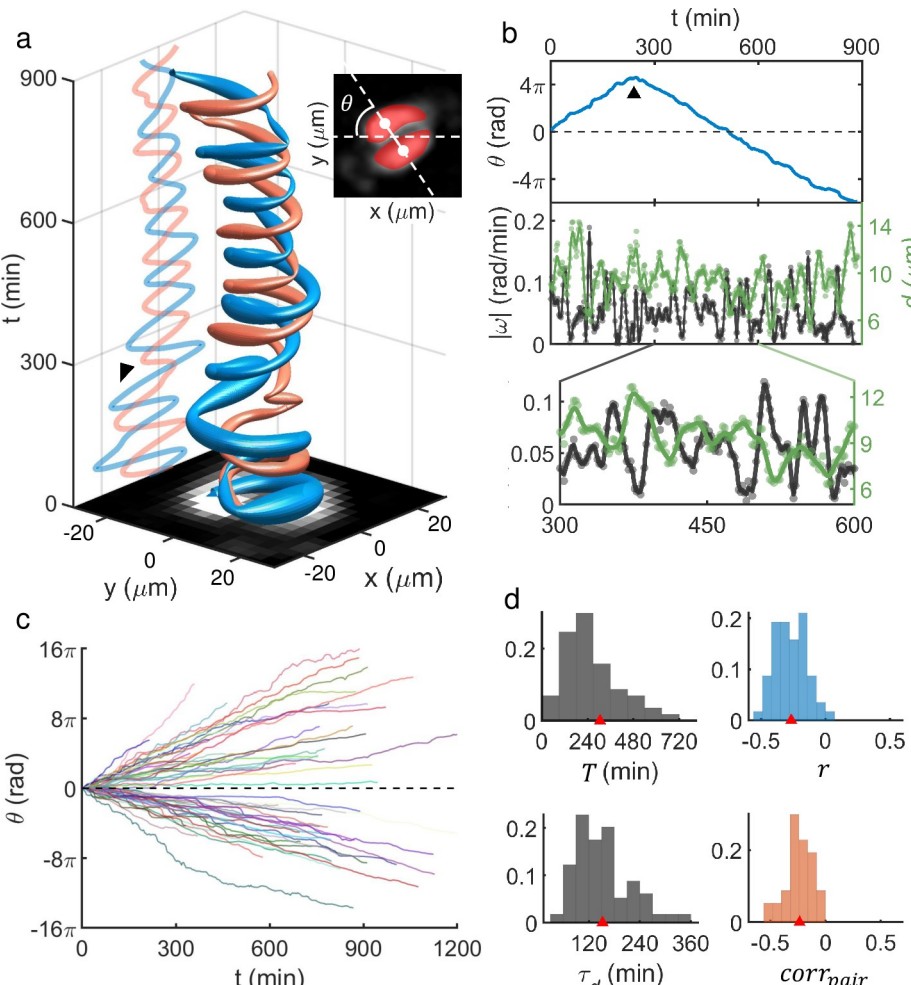

**Fig 2. Robustly rotating MDA-MB-231 doublets and their characteristics.** (a) Space-time plot showing trajectories of two cells in contact (see inset) forming a pair. Again, 'tube' thickness represents instantaneous speed. The arrow marks an abrupt reversal turn. (b) Unwrapped rotation angle $\theta$ (top), angular speed $|\omega|$, and cell's (centroid) displacement $d$ from the doublet's center (middle and bottom) as a function of time [dots: data points; solid lines: spline (3rd order polynomial function) fits to the data points]. $\theta$ was measured in the moving frame of the pair's center. (c) The unwrapped rotation angle of doublets vs. time. Total 57 doublets were tracked. All abrupt, intermittent, reverse-turns were flipped to keep the initial rotation chirality to emphasize the variation of rotation speed among the ensemble. (d) PDFs showing a significant degree of heterogeneities over the ensemble of analyzed doublets [from upper left to lower right, mean rotation period $T$ (305 ± 371 min), Pearson correlation coefficient $r$ between $|\omega|$ and $d$ (-0.3 ± 0.1), modulation period $\tau_d$ (153 ± 64 min) which we obtained by finding the frequency giving the maximum intensity in the Fourier transform of $d$, and pair-correlation $corr_{pair}$ which we defined as the temporal mean of the dot product between unit velocity vectors of each cell (-0.2 ± 0.1); all are given as (mean ± std)].

movement as shown by the example in Fig 2A. This doublet rotated in a highly periodic fashion about 7 times in 900 minutes, during which one abrupt reverse-turn occurred between $t = 240$ min and 300 (marked by an arrow in Fig 2A; also see the top frame of Fig 2B). [Abrupt reverse-turns were common but intermittent (see S3 Fig for the frequency of reverse-turn events)]. Meanwhile, there was strong anti-correlation (Pearson correlation coefficient $r = -0.3$ over the given time duration) between each cell's displacement $d$ from the center of doublet and its instantaneous angular speed $|\omega|$, which is shown in the second frame of Fig 2B, with an inset of the blown-up image shown below. In other words, the rotation slowed down as both

cells maximally stretched. Thus, the doublet rotation must not be viewed as a simple rigid body rotation but more as a continuous position swapping between two contacting cells.

The observed MDA-MB-231 pair-rotation was a robust phenomenon (see Fig 2C and S3 Movie): 57 pairs (pooled from three different cultures) maintained their integrities for more than 212 min, which was the minimum time duration of tracked doublets. For some doublets, the rotation lasted as long as 1324 min beyond which they replicated or separated. Note that the slope of unwrapped angle $\theta$ vs. time $t$ plot in Fig 2C varies significantly from one to the others. In fact, the heterogeneity in the pool of analyzed doublets could be quantified with a number of different measurements. Mean rotation period $T$ for each doublet was defined to be $T = 2\pi/\overline{|\omega|}$ and its broad probability distribution function (PDF) is shown in the upper left frame of Fig 2D. Also, we could characterize the oscillatory behavior in $d$ by calculating the peak position time $\tau_d$ of the Fourier transform of $d$. The PDF of $\tau_d$ is shown in the lower-left position of Fig 2D, which is also broadly distributed. We noted that the ensemble mean of $T$ (= 305 min) was very close to twice as much as the mean of $\tau_d$ (= 153 min) confirming that two stretches (and contractions) had occurred during one complete rotation. $corr_{pair}$, which we defined as the temporal mean of $\emptyset(t) = \hat{e}_1(t) \cdot \hat{e}_2(t)$, as well as the Pearson correlation coefficient $r$ exhibited a broad PDF as shown in the lower (upper) right frame of Fig 2D, respectively.

## Description of cell-quadruplets

The rotational behavior of MDA-MB-231 doublets was conserved even as they grew to a cell-quadruplet, yet with additional complexity and intrigue that were incurred by having multiple neighbors (see Fig 3). Four cells forming a quadruplet cluster could sustain a steady 'unimpaired' rotation (which we refer to as a rotation with no change in cyclic positional order) but typically only for some short duration of time: See, for example, the dynamics during the time window $t = 260 \sim 440$ min which is highlighted as cyan shading in Fig 3A and 3B; and its matching sequence of snapshots is given in Fig 3C, top row. This counterclockwise unimpaired rotation of four (pseudo-color coded) cells during $t = 260 \sim 440$ min was robust with their phase ordering (red, yellow, blue, violet) maintained, but was not a rigid body rotation as their relative distances to the centroid of the quadruplet significantly varied (see green lines in Fig 3B). Typically, the unimpaired rotation was interdigitated by frequent position swapping events: A good exemplary time window is highlighted as red shading in Fig 3A and 3B, and its matching sequence of snapshots is given in Fig 3C, bottom row. During the time window of $t = 60 \sim 240$ min, two events of pairwise position swapping took place (first, between the purple and the yellow, and second, between the blue and the yellow), which are marked by two dashed black circles in Fig 3B (top). The swapping events were rather quick in the sense that the angular speed $|\omega|$ (black lines) peaked very sharply as shown in Fig 3B and the peaks' bandwidths were a mere fraction of the typical period (~ 640 min) of quadruplet rotation. In addition, there were concurrent changes in $d$ but at a much slower time scale, suggesting that there were some significant cell-stretching and contraction dynamics involved during and beyond the abrupt change in $\theta$ associated with a swapping action. Thus, we may as well view this swapping action as a 'mixing' process within a rotating cell-quadruplet. Note that in the aftermath of the two swapping events, the initial clockwise unimpaired rotation (for example, during $t = 0 \sim 70$ min) of the cluster became counterclockwise (for $t \gtrsim 240$ min).

The discussed MDA-MB-231 quadruplet rotation phenomenon was universal (see S4 Movie), and once they formed a cluster, they rarely disintegrated. However, as it was the case for the rotating doublets, the analyzed quadruplets exhibited quite heterogeneous physical properties. Fig 3D shows temporal evolution of mean rotation angle (phase) $\langle\theta\rangle$, which was

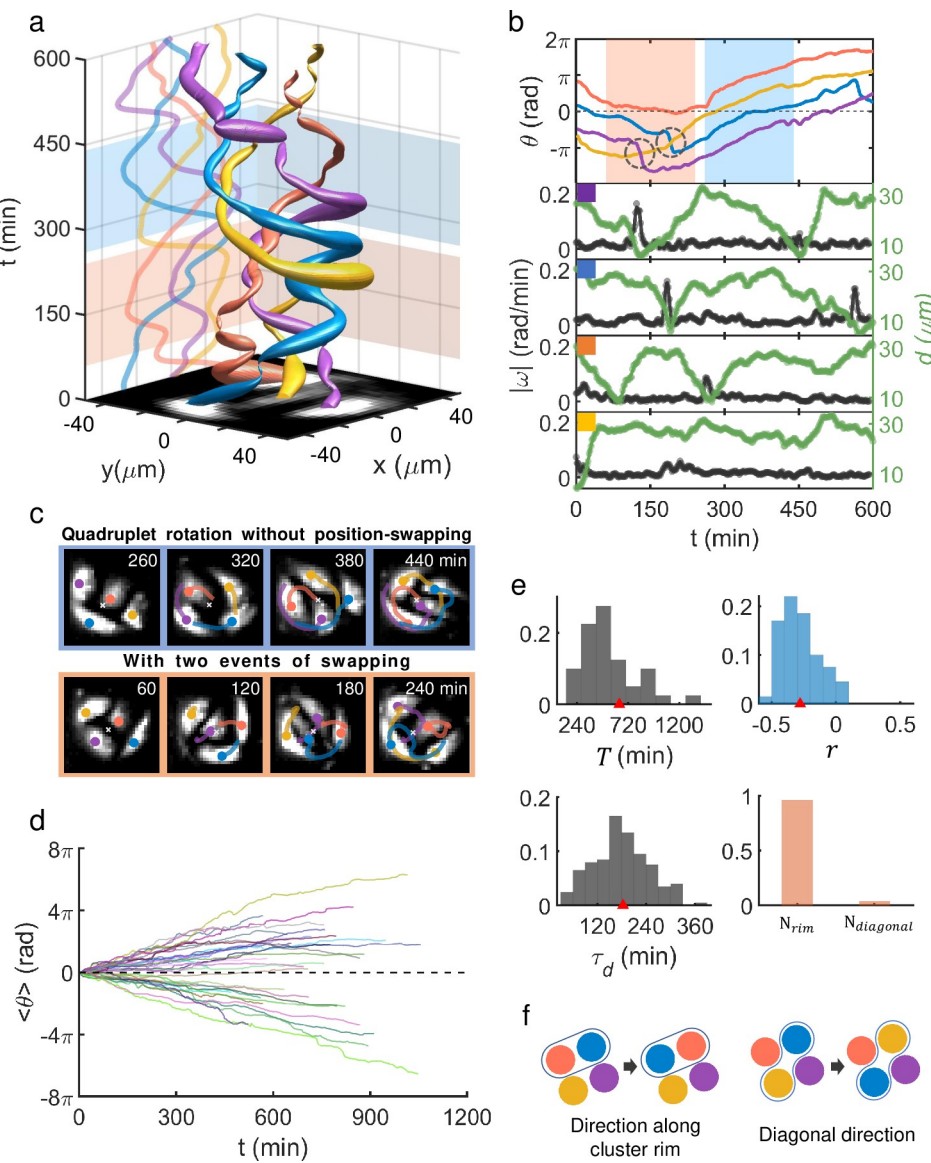

**Fig 3. Rotation of MDA-MB-231 quadruplet and intermittent position-swapping events.** (a) Space-time plot showing trajectories of four cells in contact forming a cohesively rotating quadruplet. (b) Unwrapped rotation angles for the cells [colors are matching with those of 'tubes' in (a) as well as disks and lines in (c)], angular speed $|\omega|$ (black color), and each cell's centroid displacement $d$ (green color) from the quadruplet's center as a function of time. The cyan shadow marks the time range where the quadruplet executes a robust unimpaired rotation, while the red shadow marks the range, in which two-position swapping events marked by dashed circles occurred. $\theta$ was measured in the moving frame of the quadruplet's center. (c) Snapshot images (box size of 94 $\mu m$ × 94 $\mu m$) of the quadruplet and their moving trajectories during an unimpaired counterclockwise rotation (top row), and those during a complex rotation which includes two cell position swapping events (bottom row). (d) Mean rotation angle (phase) $\langle\theta\rangle$ over four constituent cells of each quadruplet as a function of time. 40 quadruplets were tracked and shown. (e) Quadruplet ensemble PDFs (n = 40): (from upper left to lower right) mean rotation period $T$ (638 ± 448 min), the Pearson correlation coefficient $r$ between $|\omega|$ and $d$ for each cell (-0.3 ± 0.2), modulation period $\tau_d$ of $d$ (183 ± 75 min) which we defined identically to the doublets' $\tau_d$; inverse of the frequency with maximum intensity of the Fourier transform of each cell's $d$ within quadruplets, and lastly the swapping-type [$N_{rim}$ stands for the case in which the swapping occurs between two nearest neighbors along the quadruplet's rim and $N_{diagonal}$ is for the case of swapping between diagonally positioned neighbors]. All numerical values are given as (mean ± std). (f) Illustration of two different types of swapping events.

calculated by averaging the instantaneous rotation angles of the four constituent cells in each quadruplet, of all tracked quadruplets. Again, similar to the doublets of Fig 2C, all global reverse turns were unfolded to straighten out the function $\langle\theta(t)\rangle$, more or less to a line, so that the overall slope of each line represents the average angular speed of its matching quadruplet. Position swapping events created jitters in the line of $\langle\theta(t)\rangle$ and caused phase lags; thus, they somewhat slowed down the overall rotation speed. Fig 3D well illustrates the wide range of slopes (i.e., angular speeds) that MDA-MB-231 quadruplets could support. Subsequently, the PDFs of mean rotation period $T$ given in the upper-left frame of Fig 3E show a broad spectrum. Note that its ensemble mean value of 638 min was much larger than the ensemble mean of rotation period of the doublets, which was 305 min: This was of course natural since cells in a quadruplet must travel a longer distance than those in a doublet to complete a turn as the cluster (area) size had approximately doubled; furthermore, once again there were many phase-delaying position swapping events for the quadruplets. The other PDFs in Fig 3C all consistently show a broad spectrum. Interestingly, the mean value of $\tau_d$ (183 ± 75 min) and that of $r$ (-0.3 ± 0.2) both matched to those of doublets discussed earlier (153 ± 64 min and -0.3 ± 0.1, respectively) very closely. This could be an indication that the observed swapping events might not be a stochastic process but related to the innate nature of contacting doublets making rotational movement. Incidentally, we find the observed position-swapping events within MDA-MB-231 cell quadruplets almost exclusively took place not along 'diagonal direction' but along 'normal directions' [see Fig 3E (lower right frame) and the schematic illustrations shown in Fig 3F].

## Simulated cell-population vs. freely crawling cells

Most of the features from our experimental observations could be faithfully recapitulated by a CPM of active cells which can generate self-propulsive movement [32–34]. As we will demonstrate, self-propulsion strength factor $S$ and interfacial energy $E$ were indeed key parameters governing the collective behavior of confluent population as well as the coordinated movement of cell-doublets (see Methods for more details about other parameter values used for the simulation). Before we present a complete two-dimensional phase-diagram spanned by $S$ and $E$ showing various modes of cell motility, we first discuss our specific simulation results that closely matched those that we observed in experiments. The particular choice of $S$ and $E$ used for the simulation was validated by a "self-consistency check" guided by our experimental data. In other words, almost all physical properties of rotating doublets, mixing cell-quadruplets, and non-flocking, super-diffusive migration of cell-population, were recapitulated successfully with a single set of $S$ and $E$.

Fig 4, which was created based on CPM simulations, almost exactly replicates its experimental version of Fig 1. $S = 2.8$ and $E = -65$, and all the other parameter values which we kept fixed throughout this paper are described in Methods. Greater diffusivity of the cells in confluency (Fig 4A) than that of freely migrating cells (Fig 4B) was indisputable. The enhanced diffusivity is also evident in Fig 4C and 4D: For the time range of around $t \lesssim 10^{2.5}$ (or $10^3$) min, the model cells were super-diffusive (see Fig 4C and 4D) and, more importantly, the diffusion exponent $\alpha$ was significantly larger for the cells in confluency. Along with the auto-correlation functions $c_{auto}$ shown in Fig 4E, these data were consistent with the increased average persistence time $\langle\tau_2\rangle$, instantaneous velocity $\langle v\rangle$, and persistence length $\langle l\rangle$ for the cells in confluent population [$\langle\tau_{2,pop}\rangle = 42 \pm 8$ min vs. $\langle\tau_{2,free}\rangle = 26 \pm 4$ min, migration speed $\langle v_{pop}\rangle = 0.7$ $\mu m$/min vs. $\langle v_{free}\rangle = 0.4$ $\mu m$/min, $\langle l_{pop}\rangle = 27 \pm 6$ $\mu m$ vs. $\langle l_{free}\rangle = 11 \pm 2$ $\mu m$]. At this point, we should point out that the two calculated $\langle\tau_2\rangle$ values from the experiment and the simulation (for both freely crawling and in confluency) differed by a factor of around 3. This difference

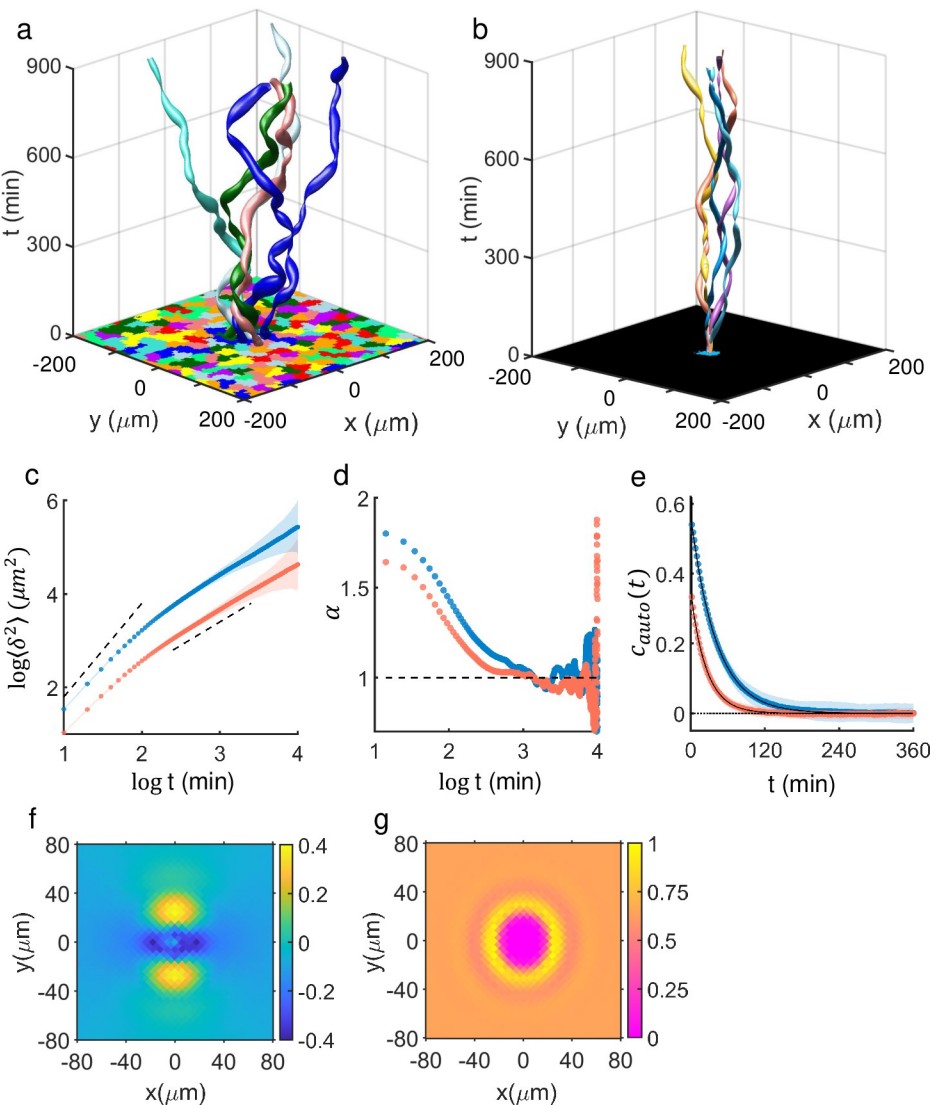

**Fig 4. CPM computer simulation comparing motile properties within a dense population and those of freely crawling cells.** (a) The confluent population of model (arbitrarily colored) cells and exemplary space-time trajectories of 6 cells, which were adjacent to each other at $t = 0$ (number of simulated cells = 990). (b) Superposition of space-time trajectories of 6 freely migrating model cells with no cell-cell interaction; all start out from the origin at $t = 0$. (c) MSD vs. time plots (dots: mean, shadow: std); dashed lines represent the slope of 2 and 1, respectively. (d) The plot of diffusion exponent $\alpha$ as function of time. $\alpha$ was obtained by calculating slope of MSD between two consecutive data points shown in (c). (e) Mean auto-correlation $c_{auto}$ of the unit tangent vector along the trajectory is marked as circles. The overlaid solid lines are a fit of two-tier exponential function $C(t) = Ae^{-t/\tau_1} + Be^{-t/\tau_2}$: blue line, $(\tau_1, \tau_2) = (0.10$ min, $40.95$ min) and $(A, B) = (0.43, 0.57)$, and for the red line, $(\tau_1, \tau_2) = (0.49$ min, $25.40$ min) and $(A, B) = (0.65, 0.35)$. R-squared values were ~ 1.00 (rounded at third decimal place) for both cases. The MSDs and the auto-correlation functions were calculated with data points separated by 1 MCS. (f) (Temporal and ensemble) average spatial correlation map of $c_{cross}$ of unit tangent vectors pointing the instantaneous directions of movement. (g) Normalized average density map of neighboring cells in a confluent population. The oval area (aspect ratio of 0.8) represents an average shape of reference cell moving towards the positive y-axis. For (c)-(e), blue (red) represents cells within the confluent cell population (freely crawling cells), and all mean values were computed on n = 200 randomly chosen cells (trials). The values mapped in (f) and (g) reflect temporal (total simulation time of $10^4$ MCS) as well as ensemble average (n = 990).

could have originated from a few different sources. First of all, for the given set of parameter values that we chose the model cells turned out to be somewhat stiffer than the real MDA-MB-231 cells: This is clear when Fig 4G is compared with its experimental counterpart of Fig 1G; the oval shape (i.e., ensemble and time-averaged cell shape) is more elongated in Fig 1G. Second of all, there was an ambiguity in setting a Monte Carlo step (MCS) to an exact physical time, an issue which we discussed in Methods.

Fig 4F displays a velocity-velocity correlation map for the model cells in confluency, and it clearly shows that the range of cell-cell interaction is very short, about a cell diameter ($\sim 30$ $\mu m$), which is consistent with the experiment in Fig 1F. Once again, the map confirms that neighboring cells on both (left and right) sides of a reference cell tended to move along the opposite direction to which the reference cell is heading. These results are an indication of uncaging and rearrangements of the cells in confluency, and reminiscent of the position swapping observed in MDA-MB-231 cell-doublets and cell-quadruplets.

## Simulated cell-doublets

Fig 5 illuminates simulation results of CPM doublets, for which we used the same set of parameter values used in Fig 4 ($S = 2.8$, $E = -65$). Several experimental features of doublets were reproduced remarkably well: robust rotation of doublets having some occasional reversal turns [see Fig 5A and 5B (top panel)]; anti-correlation between angular speed $|\omega|$ and the cell-center displacement $d$ from the doublet's center (lower panels in Fig 5B); robustness of the rotation (Fig 5C); and the fact that mean modulation period $\tau_d$ of the cell-center displacement was approximately half the mean rotation period $T$ (Fig 5D). The spread of the unwrapped $\theta$ lines shown in Fig 5C reflects a different number of intermittent reverse-turns for the given time duration. The ensemble mean of $T$ (= 213 ± 13 min) and $\tau_d$ (= 99 ± 22 min) for the CPM doublets are 0.70 and 0.65 times of those of the MDA-MB-231 cell pairs, respectively, and this discrepancy might be attributed to the inaccuracy in the parameter values, in particular, associated with the CPM Hamiltonian energy part which determines the overall shape and the softness of individual cells. On the other hand, the mean of Pearson correlation coefficient $r$ (= -0.2 ± 0.02) and $corr_{pair}$ (= -0.3 ± 0.03) from the simulation shown in Fig 5D, both fall well within the measured range of $r$ (-0.4 ~ -0.2) and that of $corr_{pair}$ (-0.3 ~ -0.1) of the MDA-MB-231 doublets quantified in Fig 2.

## Simulated cell-quadruplets

The quadruplet dynamics of MDA-MB-231 cells, including intermittent position swapping behavior, were also well captured by CPM simulation as shown in Fig 6A. Rotation of four cohesive cells was quite robust [for example, see the exemplary time window highlighted in cyan in Fig 6B and 6C (top row)]; however, some intermittent position swapping events [see the time window highlighted in red in Fig 6B and 6C (bottom row)] were also visible as in experimental MDA-MB-231 quadruplets of Fig 4. Also, the anti-correlation between $|\omega|$ and $d$ is also evident in Fig 6B. Fig 6D illustrates the ensemble of the $\langle\theta(t)\rangle$ lines. Again, position swapping events brought noisy jitters in the lines of $\langle\theta(t)\rangle$ and caused phase lags, therefore, the overall slope of $\langle\theta(t)\rangle$ could vary depending on the number of position-swapping events for the given time duration. Fig 6E summarizes ensemble statistics of three measures ($r$, $T$, $\tau_d$) which we also used for the quantification of the experimental data in Fig 3E. The mean value of $r$ = -0.3 (for the simulated quadruplets) is well within the broad range (-0.5 ~ -0.1) of $r$ covered by the MDA-MB-231 quadruplet ensemble in the experiment. We note that the mean rotation period $T$ = 282 min of the simulated quadruplets is about 1.32 times longer than the average period $T$ = 213 min of the simulated doublet, while $T$ = 638 min of MDA-MB-231

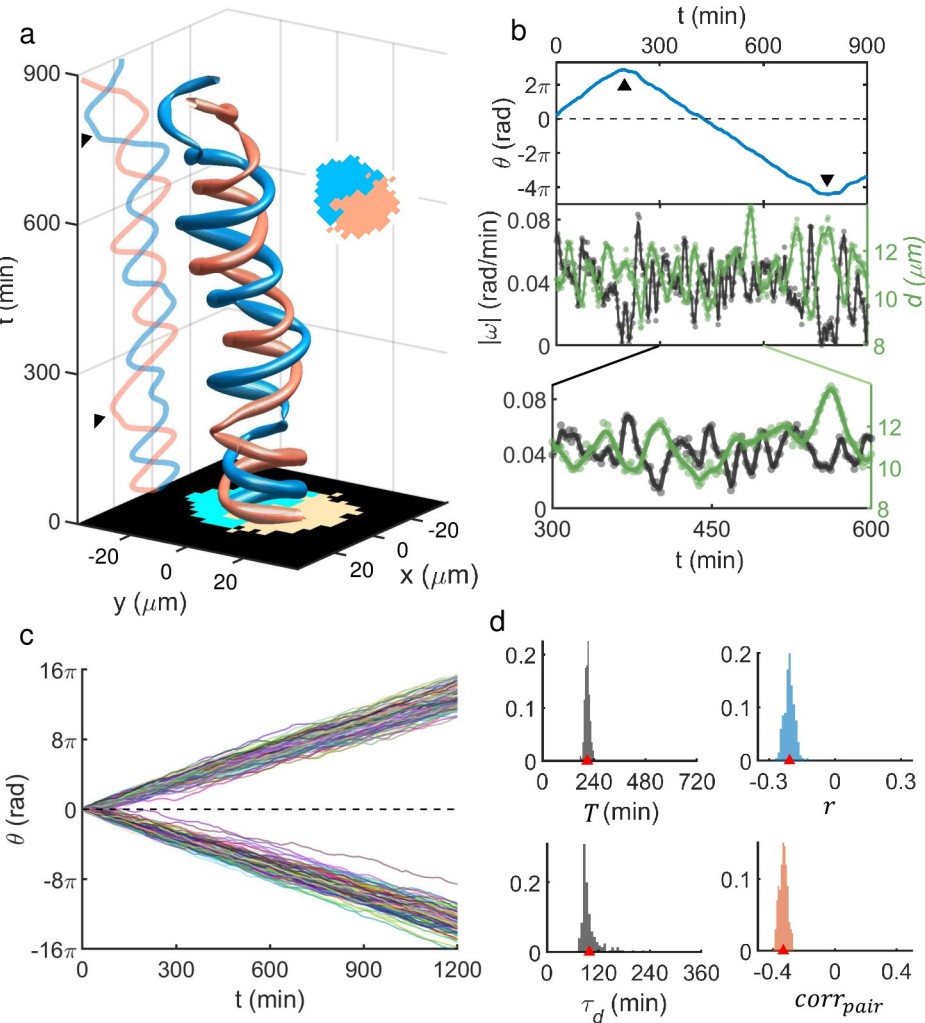

**Fig 5. Dynamic and statistical properties of rotating cell doublets in CPM simulation.** (a) Space-time trajectories of an exemplary doublet. The tube thickness represents the size of instantaneous speed, and two arrows mark the position of doublet reverse-turns. (b) Unwrapped rotation angle of constituent cells with respect to doublet centroid (upper panel), $|\omega|$ and $d$ (lower panels) vs. time. Dots represent actual data points acquired at each MCS and solid lines are spline fits with the same method used for Figs 2 and 3. Time-points where reverse-turns occurred were marked with arrows. (c) Unwrapped rotation angle vs. time. As in the experimental analysis, abrupt reverse-turns were flipped to keep the initial rotation chirality to better visualize the overall rate of angular rotation. A total of 200 doublets were tracked. (d) Various PDFs over the ensemble of analyzed doublets [from upper left to lower right, mean rotation period $T$ (213 ± 13 min), Pearson correlation coefficient $r$ between $|\omega|$ and $d$ (-0.2 ± 0.02), modulation period $\tau_d$ of $d$ (99 ± 22 min) and pair-correlation $corr_{pair}$ (-0.3 ± 0.03)]. All were given as (mean ± std).

quadruplet is about 2.09 times longer than the period $T$ = 305 min of MDA-MB-231 doublet. This discrepancy was mainly due to the fact that MDA-MB-231 cells experienced position swapping events more frequently than the CPM cells. Another factor for the discrepancy could be that the actual cells were softer and more ramified than the model cells; we hypothesize that such properties could result in more severe shape deformation that in turn slows down the coherently directed rotation. Finally, the bar graph in the lower right of Fig 6E confirms that the swapping between diagonal neighbors was almost non-existent as observed in the experiments.

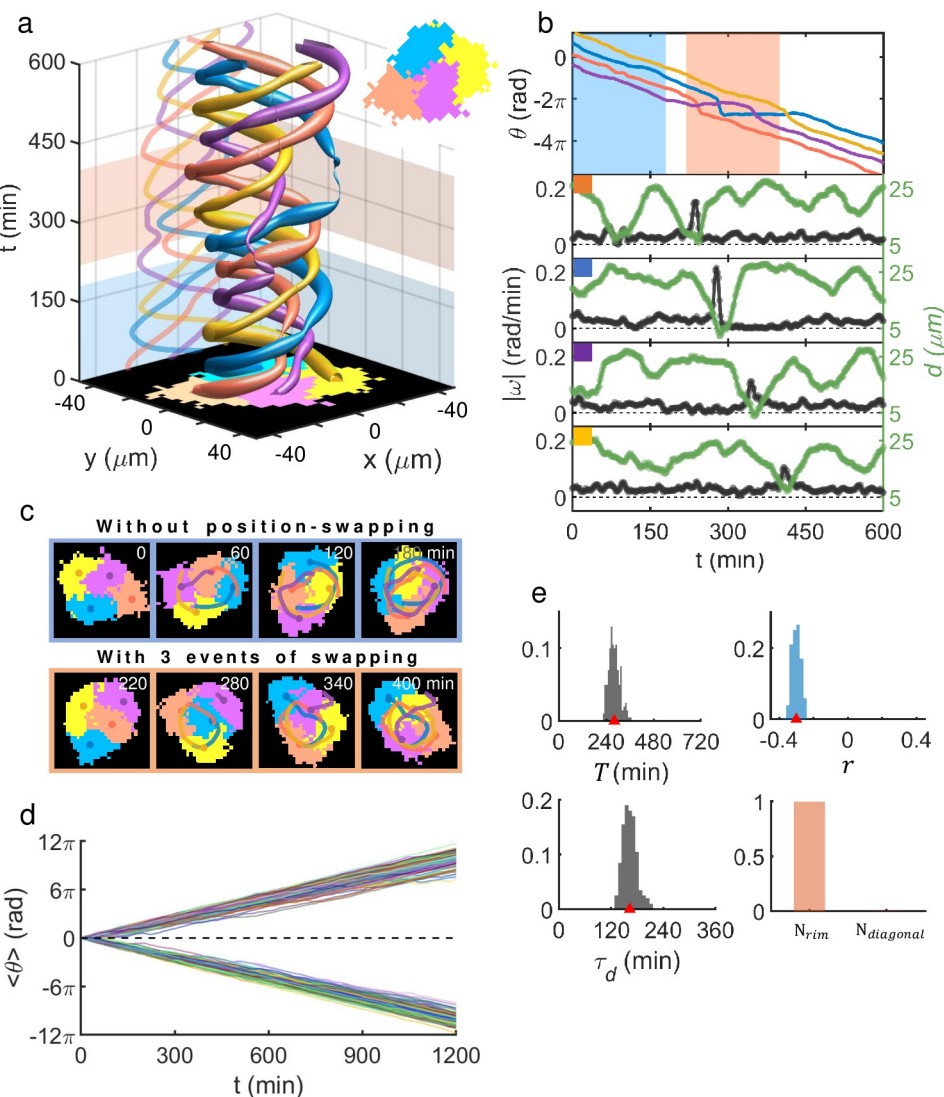

**Fig 6. Dynamic and statistical properties of rotating cell quadruplets in CPM simulation.** (a) Space-time trajectories of an exemplary quadruplet. (b) Unwrapped rotation angle of constituent cells with respect to quadruplet's centroid (upper panel), $|\omega|$ and $d$ (lower panels) vs. time. (c) Sequences of snapshots taken during the two highlighted time windows that are color-marked in Fig 6B (top panel). (d) Unwrapped rotation angle vs. time. Again, all global reverse-turns were flipped to keep the initial rotation chirality. Total of 200 quadruplets are tracked. (e) Various PDFs over the ensemble of analyzed model quadruplets: (from upper left to lower right) mean rotation period $T$ (282 ± 28 min), Pearson correlation coefficient $r$ between $|\omega|$ and $d$ (-0.3 ± 0.03), modulation period $\tau_d$ of $d$ (163 ± 16 min), and swapping-type. $N_{normal}$ ($N_{diagonal}$) stands for the case the swapping is between two normal (diagonal) axes. All were given as (mean ± std).

## Doublet phase diagram

The colormaps shown in Fig 7 summarize our CPM simulations. Fig 7A and 7B are phase diagrams spanned by $S$ and $E$ plotting for $corr_{pair}$ and $r$ of the simulated cell-doublets, respectively. In Fig 7A, negative values (blue) indicate persistent, stable rotation. Near the right bottom of the diagram $\langle corr_{pair} \rangle$ is close to zero (dark yellow); it is a region where pairs failed to maintain adhesion (see S4A Fig). Since $\langle r \rangle$ = -1 means a complete anti-correlation between $|\omega|$ and $d$, the valley with prominent sky-blue color in Fig 7B speaks for more pronounced position swapping and stretched rotation. With small $S$ and $E \sim 0$, doublets did not exhibit interaction,

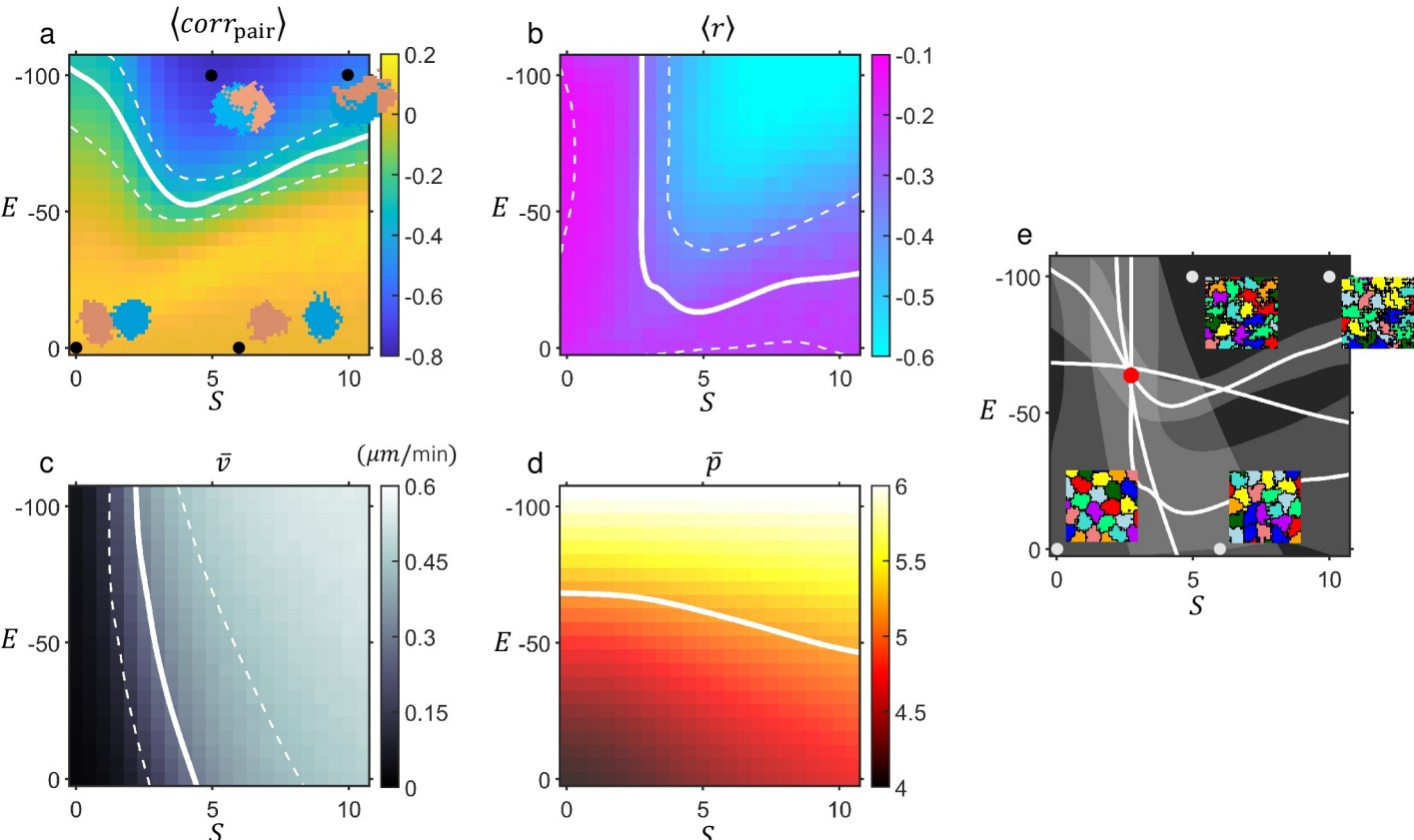

**Fig 7. Phase diagrams in *S–E* plane for doublet (7A and 7B) and confluent cell-population (7C and 7D) in CPM.** Phase diagrams in (a)-(d) represent heatmaps (and color bars) for 4 different measurements obtained from the simulations. (a) Doublet's mean pair-correlation $\langle corr_{pair} \rangle$ (see the caption of Fig 2 for definition). (b) Mean Pearson correlation coefficient $\langle r \rangle$ between doublets' angular speed $\omega$ and rotation radius $d$. (c) (Temporal and ensemble) average of moving speed $\bar{v}$ of cells within confluent population. (d) (Temporal and ensemble) average of shape index $\bar{p}$ of the simulated cells within confluent population whose distribution is shown in Fig 1H. Particularly, (a) illuminates the doublets' tendency of rotational motion: the value ranges from -1 (stable rotation) to 1 (co-movement), while the doublets' 'stretched rotation' is elucidated in (b) with negative values closer to -1 meaning more severely stretched rotation. White solid lines for all four diagrams in (a)-(d) represent level-curves for experimentally obtained corresponding values, while white dotted lines are the bounds set by the experimental standard deviation from the mean (for (d), dotted lines are out of range of the diagram). For example, in (a) the lines correspond to the mean (and the deviation from the mean) of $\langle corr_{pair} \rangle$ whose ensemble distribution is shown in Fig 2D—lower right panel. Similarly, for (b) the white lines represent the data from Fig 2D—upper right panel. Doublet snapshot images are illustrated for 4 different sets of (*S, E*)s, marked by black dots in (a). (e) All 4 level curves (of the means) of (a)-(d) were collected and superimposed with corresponding standard deviations given as shades: Red dot, located very near to the crossing point, marks the set of parameter values (*S* = 2.8, *E* = -65), which has been used for all exemplary CPM simulation given in Figs 4–6. Snapshots of 4 exemplary populations are given for 4 different sets of (*S, E*)s which are marked by gray dots.

failing to make a turn (S5 Movie shows the process at *S* = 0, *E* = 0). As *S* became large ($\gtrsim$ 5) with *E* ~ 0, doublets separated (S6 Movie at *S* = 6, *E* = 0). Doublet rotation was visible as *E* became more negative while *S* being non-negligible (S7 Movie at *S* = 5, *E* = -100). The threshold value of *S* for rotation tended to decrease slightly as *E* became more negative for the region *S* $\lesssim$ 5 (see S4B Fig). In the upper right corner of the phase diagram, more stretched rotation (S8 Movie at *S* = 10, *E* = -100) were visible.

## Confluent population phase diagram

Fig 7C and 7D are phase diagrams for the (temporal and ensemble) average absolute velocity $\bar{v} = \overline{|\overrightarrow{r}(t + t_0) - \overrightarrow{r}(t)|}/t_0$ (with $t_0 = 180$ min), and average shape-index $\bar{p}$ of cells within confluent population. Among the variables which we could measure in experiment, we chose $\bar{v}$ and $\bar{p}$ as two key properties of the cells within confluent population. $\bar{v}$ was selected as a good

measure of the diffusivity of the cells, and $\bar{p}$ was chosen since it has emerged in recent studies as an important structural signature for the migratory transition of a confluent tissue [5,7]. Clearly, $S$ was critical for bringing a jammed population (S5 Movie) to an unjammed state for the entire range of $E$, while decreasing $E$ facilitated the unjamming for the case of a small (~ 2) $S$ (Figs 7C and S5A). For the migratory state around $S \sim 5$ and $E \sim 0$ (i.e., no adhesiveness), diffusion exponent $\alpha$ calculated at a long time-scale showed super-diffusivity with greater than 1 (S5A Fig), and concurrent rise in the average size of cohesively moving cell-clusters was visible (S5B Fig and S6 Movie). As $E$ decreased, cells migrated more diffusively and individually (S7 Movie), while interfaces became more ragged (Fig 7D). Finally, all four level curves shown in Fig 7A–7D were put together in Fig 7E, and they almost meet together at a common point (red dot, $S = 2.8$, $E = -65$), whose values of $S$ and $E$ were used for all CPM simulations given in Figs 4–6.

Increasing the self-propulsion parameter $S$, with a fixed value of $E = -65$ corresponding to the red dot in Fig 7E, intermittent co-movement (flocking) of the doublets became more frequent (S6 Fig). As it happened, the instantaneous angular velocity decreased while linear velocity increased. The time-fraction of doublets' flocking as a function of $S$ indicates that there is a continuous transition between the rotation and flocking. The time-shared mixed behavior is somewhat similar to the mixed motility (running, rotation, and random individual motion) mode of larger cell-clusters under the gradient of chemoattractant as discussed in [35,36]. Subsequently, we postulated that the cells in a confluent population might possess the two different modes of cell motion which were supported by doublets: namely, the flocking and rotation. Indeed, the velocity correlation maps (Figs 1F and 4F) strongly supported that idea: positive (negative) correlation along the migration (perpendicular) axis is evident. Besides, the correlation length along the migration axis (y-axis) of the cells within confluent population also revealed a similar transition as a function of $S$, which eventually led to a correlation length comparable to a single-cell's diameter (S7A Fig). The negative correlation along the x-axis, reminiscent of rotation with adjacent neighbor, existed over a wide range of $S$ (see S7B Fig).

## Discussion

Super-diffusivity of freely crawling animal cells (on 2D substrate) is not an uncommon phenomenon, being supported by many different types of cells, and its underlying mechanism has been explored in numerous previous studies [2,37–40]. One of the essential properties for the super-diffusivity is the self-propelling force, generated by polymerization of actins at the moving front, and depending on the strength (and time duration) of this force (as well as other factors influencing cortical actin dynamics) the phenotypic cell motility would be rather different. What is surprising in this work is that the diffusion exponent $\alpha$ is larger for the cells in confluent populations compared to that of freely crawling cells.

The observed phenomena can be realized with cells that have a proper degree of directional persistence and are just sticky enough to hold on to each other. For the chosen set of $(S, E)$, the cells forming a doublet rotate about each other. But in other regime the cells can be co-moving (flocking). The transition between the two different modes is continuous. The enhanced directional persistence and super-diffusivity in confluency is a consequence of acquiring a small but finite velocity-velocity correlation length along the direction of instantaneous moving direction. The small correlation length along the y-axis is a reminiscence of the co-moving cells in a doublet forming a "two-cart vehicle". On average, the cell motility in confluency can be viewed as a random mixture of cell doublets co-moving or rotating together in a time-shared manner.

Cell-doublets had a dramatically different motility from that of a freely moving single cell: The doublets rotated periodically without much drift. The doublet-rotation must be a

cooperative work of the constituent cells, since isolated MDA-MB-231 cells do not exhibit rotation by themselves although there are some cells exhibiting intrinsic chiral rotation [41]. We note that similar coherent angular motion was reported earlier with human breast epithelial cells from mammoplasty and the phenomenon was discussed in connection to acini formation [42]: The authors found that blocking E-cadherin function naturally disrupts cell-to-cell adhesion to stop the rotation and that the doublet rotation involves non-trivial cortical actin dynamics. Incidentally, MDA-MB-231 cell lines are also derived from human breast epithelial tumors. So, the doublet rotation that we observed could be a salient feature of breast epithelial cells.

Rotating epithelial cell-clusters within geometric confinements were reported earlier [43,44], in which the coherence of the rotation was controlled by the size of the confinement. The authors found coherent cell rotation became disrupted in large confinements as the cells in the middle initiated an instability throughout the whole cluster [44]. In another study, freely migrating lymphocytic cell-cluster under the influence of chemical gradient exhibited intermittent rigid-body rotation amid persistent movement and directionless individual cell movements [35]. Incidentally, the frequent position-swapping between the rim-cells (or leader cells) and core-cells during the rotation is rather analogous to the cell position-swapping discussed in this paper, although in our case there existed neither extrinsic chemotactic influence nor cell-cell heterogeneity (i.e., rim- vs. core-cells). A self-propelled agent-based model [36], implementing each cell's density-dependent self-propulsion, elucidated the heterogeneous propulsion between the rim and core. The coupling between the cells of these subdivisions was found to be critical as for generating rotational behavior of a cluster.

The 'position swapping' event in the quadruplet motility that we discussed in this paper was somewhat analogues to the differential rim-core motility of a cell cluster discussed in [36]. That is, for the case of cell quadruplet rotation, the "rim-bulk graded motility" might apply transiently when there comes a position-swapping event, in which 4 cells forming a rim-only state produces a 3-cell rim and 1-cell bulk state, which is a reminiscence of frustrated state where the cell in the core drags and slows the 3 cells in the rim. However, in our case there was no pre-assigned, space-dependent graded motility.

Investigations on cell doublet rotation were conducted earlier via phase-field model simulations of doublets under confinements [45,46]. Studying the effect of different polarity mechanisms on rotation demonstrated that it can be generated either by front-front inhibition (mechanism avoiding front-front adhesion) or by polarity-velocity alignment (tendency of cell aligning its polarity to instantaneous migration velocity) [45]. As an extension of this model, implementing intercellular diffusion of inhibitory component at the contact site, confinement-free rotation of adhesive cell-doublet was shown [46]: Here, the rotation was viewed as a consequence of continuous 'walk-past' movement, crawling in the direction away from the contact while maintaining cell-cell adhesion. Fundamentally, these models employed mechanisms in which cells adhered to their partner differentially depending on the latter's cellular polarity. On the other hand, our current CPM model had no pre-built-in elements reflecting how single cell (internal) properties change upon cell-cell contact or depend on local cell density. In fact, we had no experimental data to address such issues for the MDA-MB-231 cells and incorporate them realistically into our CPM. The same was true for cell polarization: Simply, we do not know if there exists any direct polarization-polarization interaction among neighbors in our MDA-MB-231 populations.

In our CPM model, neighboring polarization vectors interacted only indirectly by the cell population dynamics, as the neighboring cells in contact could alter the cell shapes each other. For example, the interaction between two polarity vectors of a cell doublet is mediated by the interface between two neighboring cells (see Fig 8): Initially, there is a pair of initially

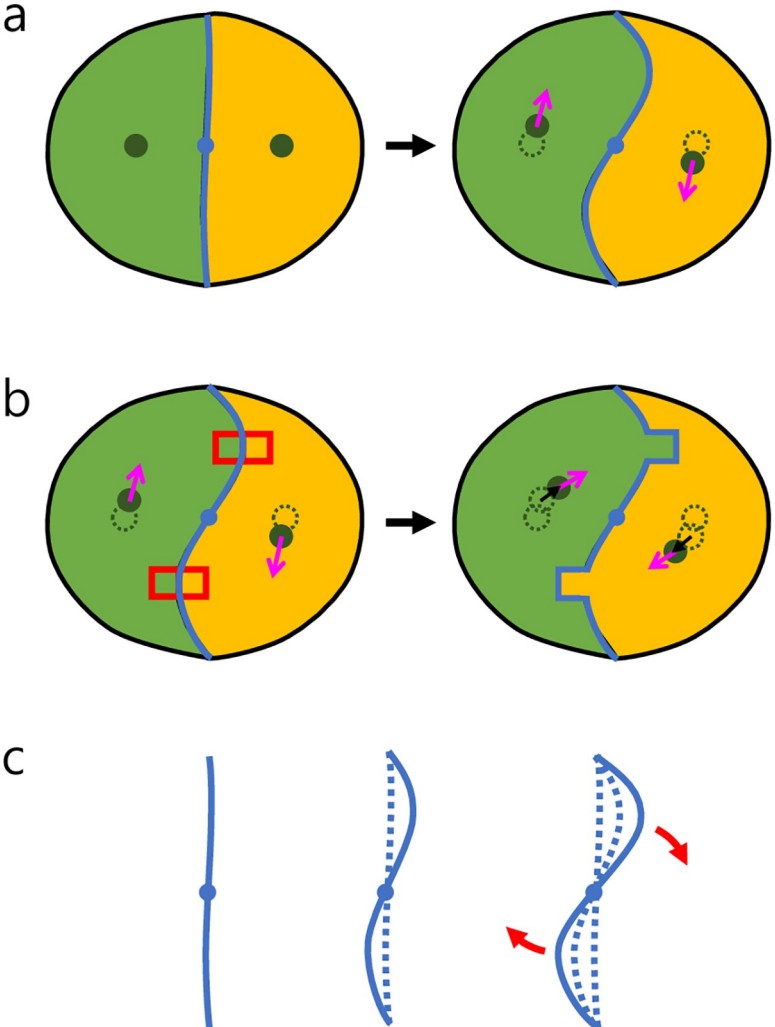

**Fig 8. Schematic illustrating how a small-amplitude deformation of cell-cell interface could be amplified by cell polarity.** (a) (Left): Initially, two mirror-symmetric, adherent cells form a straight interface; then, (Right): A simple harmonic fluctuation sets in as a random fluctuation, which deforms the cells, thereby, moves the centroids (marked by filled green circles) to initialize a pair of polarity vectors $\hat{p}$s (pink arrows) [In our model, the initial $\hat{p}$s are nothing but the normalized initial displacement vectors $\Delta x$ of the centroids]. (b) (Left): We consider CPM updates for the pixels within the two (red) highlighted boxes, considering only the 'polarization part' of our CPM energetics; (Right) The result of the pixel updates will be (probabilistically) highly like to be the image as depicted on the right [the yellow (green) pixel replaced by the green (yellow) within the red box on top (bottom)]. Consequently, a new $\Delta x$ is created (for each cell) towards which the next $\hat{p}$ will be attracted. (c) As the CPM update repeats over the time along the whole interface, the initial small-amplitude harmonic fluctuation will grow and rotate in the clockwise direction. Of course, this polarity driven (pure) rotation will compete with 'curvature-smoothing' energetic components in the CPM. In fact, the competition seems to make the actual rotation not like a rigid-body rotation but more like a periodic sequence of position swapping events as discussed in Fig 5.

unpolarized, mirror-symmetric, adherent cells; then, imagine their interface starts to deform due to random fluctuation (a basic harmonic mode is assumed); with that, their shapes (centroids) also change (move); consequently, the cells acquire polarity vectors (Fig 8A); the polarity vectors then enhance the interfacial deformation to initiate a rotation via the CPM energetics part incorporating a polarity-velocity alignment tendency (Fig 8B); the same energetics keeps amplifying the harmonic deformation (Fig 8C) to a steady profile and maintains

the rotation. Of course, the above sequence of events happens only for an appropriate parameter regime of $(S, E)$.

One limitation of current CPM was that it did not have a built-in contact-inhibition-loco-motion (CIL) mechanism, where CIL refers to the phenomenon in which the physical contact between (colliding) two (more more) cells initiates some chemical reactions that affect either the mechanics or the adhesive property of the cells involved. As discussed in the two earlier papers [45,46], CIL could be a critical element as important as cell mechanics (e.g., shapes), propulsion strength, cell polarity, or adhesion, as for governing the behavior of cells in contact. Unfortunately, we do not know if MDA-MB-231 cells had any significant CIL and if it they did how they work. The same was true for the cell polarization interaction. In our model, the polarization (as well as velocity) vectors interact only indirectly by the cell population dynam-ics (i.e., cell contacts): Each polarization vector changes as the cell's centroid position, which is affected by the shape of the cell that is influenced by its adjacent neighbors, moves in space. Simply, we do not know if there exists any "direct" polarization-polarization interaction among neighbors in our MDA-MB-231 populations.

Another limitation of our CPM model was that the polarization vector was not emergent from the CPM-calculated cell shape but was set externally. In other words, the polarization vector was not directly born out from the cell shape. We should indicate that there are variants of the CPM where polarization and persistence emerge from the underling actin dynamics [47,48], and therefore motility and persistence are highly shape dependent. Currently, it is not clear how critical to have the polarization vector evaluated from the cell shape directly. In the future, it would be interesting to simulate these variant CPMs that are based on realistic actin dynamics. Likewise, it will be also worthwhile to add on additional elements such as CIL and/or direct polarity-polarity interactions in the CPM to see how our current observations change.

The enhanced diffusivity of the confluent MDA-MB-231 cells in this report draws impor-tant similarities to the enhanced motility of human kidney cells expressing MOCA protein [25]. MOCA-expressing cells exhibited cluster formations due to enhanced cell-cell adhesion in contrast to the control. Higher diffusivity of the interacting MOCA-expressing cells was attributed to their correlated movement evidenced by the longer correlation length (amounts to roughly a cell-diameter) than that of the control cells. In the meantime, recently reported contact enhancement of locomotion (CEL) [49] considered active Brownian particles which switched to a ballistic mode of motility upon a collision, recapitulating the experimental obser-vation of Dictyostelium discoideum cells exhibiting greater persistence in higher cell-density. The CEL study employed cell cultures in which the cells can freely explore the free space before contacting another cell, contrary to the confluent situation of our densely packed samples. In other words, CEL does arise not by cell-cell adhesion but by intracellular changes brought by physical contacts. On the other hand, the NED was attributed to the local adhesion-mediated interaction. In any case, all of these studies clearly underscore the importance of understand-ing population dynamics as for the cell motility: Clearly, "More is different" [50].

If the NED effect were demonstrated here on MDA-MB-231 cells, we would expect it to be relevant to many different amoeba-like immune as well as tumor cells in a much broader con-text. Generally, NED property would endow efficient exploration or invasion inside a close-packed tissue environment, which may be critical for immune or metastatic cells, as well as in morphogenesis. As mentioned, the observed NED effect has a strong similarity to the CEL [49]. We suspect that CEL and NED could share common biochemical origins at their micro-scopic scale. In fact, our earlier observation on the contact interaction between a regular MDA-MB-231 cell and an enormously expanded senescent MDA-MB-231 cell clearly indi-cates that MDA-MB-231 cells prefer to move along the other cells' boundaries [51].

Although MDA-MB-231 cell lines are monoclonal, the cells are known to be quite heterogeneous in many different aspects [52–54]. Indeed, the tumor cell populations in our cell cultures exhibited a wide spectrum of diffusion exponent $\alpha$ as shown in Fig 1C. In principle, this heterogeneity in motility can be caused by extrinsic as well as intrinsic stochastic gene expressions. The simulation results in Figs 4–6 were based on a fixed set of parameters for the sake of simplicity. We have also carried out more realistic simulations by incorporating distributed values of parameter $S$ across the population of cells. Resulting MSD profiles of confluent as well as freely crawling cells became more similar to the experimental results (S8 Fig). Notably, the variation of MSD profiles of confluent population was significantly smaller than that of freely crawling cells (S8A Fig). Also, having a wide dispersal of $E$ (with a fixed $S$) for the population created no significant variation in MSD profiles; thus, the self-propulsion force $S$ (over $E$) seems to be more responsible for the broad distributions seen in the experimental results.

## Conclusion

Our bottom-up approach successfully delineated the physical behaviors of freely crawling cells, cell-doublets, cell-quadruplets to the cells within confluent population in a systematic way. More specifically, we presented the counter-intuitive experimental observation of enhanced diffusivity of MDA-MB-231 cells in confluent 2D cell-population. Also, we provided unequivocal evidence of cell-doublet rotation, which could be viewed as a continuous sequence of cell position swapping events, rather than a steady rigid-body rotation. The doublet rotation phenomenon was conserved even in cell-quadruplets, yet with an additional complexity: Unimpaired, coherent rotation of four cohesive cells was often hampered by intermittent events of pairwise cell-position swapping, each of which caused a reordering in the cyclic rotation sequence of the four cells involved. This swapping event within cell-quadruplets might be a reminiscence of sudden doublet rotation reversal. In the same light, we might view the uncaged, individual migration of MDA-MB-231 cells in confluency as an intermittent sequence of position swapping with immediate neighbors based on the observation of short spatial correlation length, and negatively correlated movement with immediate neighbors within population.

Essentially, all experimental observations made with MDA-MB-231 cells were successfully recapitulated by a CPM with a suitable choice of $S$ and $E$, which were chosen self-consistently so that cell-doublet behavior, as well as cell-population dynamics, could match experimentally measured characteristics. With the same set of parameter values, the remarkable dynamics of cell-quadruplet was also reproduced successfully. Again, what seems to be the most significant of our current CPM study is that even without having any complex CIL or direct polarity-polarity interactions, the toy model can still well reproduce all $n = 1$, $n = 2$, $n = 4$, and $n \to \infty$ phenomena rather well based on a single set of parameters $(S, E)$. Finally, we should point out that our current work did not address any detailed intracellular programs or molecular pathways regulating cell-cell adhesion or directional persistence of crawling cells, on which there have been numerous studies by others [55–58].

## Methods

### Cellular potts model description

Briefly, CPM is a lattice-based cell (or cell population) model, in which cells are defined as a simply-connected group of lattice sites, evolving in space and time based on Monte Carlo simulation with Metropolis algorithm. It has been widely used for describing various phenomena in cellular biology [59]. The update process of the model is the following: 1) we randomly pick a (uniformly distributed) lattice site and an 'neighbor' site from a local area of given radius

centered around this site. 2) The probability of the initially chosen site accepting the neighbor site's cell-type is dependent on Metropolis probability $p_{accept}$ which is a function of the change in the total Hamiltonian generated when the chosen site actually accepted the new cell-type. The $p_{accept}$ is given as 1 if the change in total Hamiltonian $\Delta H$ was negative (always accept), and decays exponentially ($e^{-\Delta H/T}$) if $\Delta H$ was positive with the temperature $T$ determining the decay rate.

In this study, the Hamiltonian includes three energy components associated with the cells' target surface areas and volumes, interfacial (e.g., cell-to-cell) adhesion, and the degree of a cell's activeness. Mathematically, $H = \lambda_{area}\sum_i(A_i - A_{target})^2 + \lambda_{surface}\sum_i(s_i - s_{target})^2 + \sum_{interface}E_{\tau_i\tau_j}(1 - \delta_{ij})$, where $A_{target}$ is the target area and $s_{target}$ is the target roughness defined as the fraction of circumference of a circle which has the same area as the given cell. Greater $s_{target}$ produces more tortuosity and ambiguity in interfaces. $\lambda$s are strengths of the energies associated with area and surface constraints which penalize deviations from the respective target values. $E_{\tau_i\tau_j}$ defines the interfacial adhesiveness between two cell-types $\tau_i$ and $\tau_j$. There is an additional term in the change of Hamiltonian $\Delta H_{motion}$ which gives bias to the migration in the direction of the 'polarity' $\overrightarrow{p}$ defined for each cell: $\Delta H_{motion} = -S\sum_i\frac{\overrightarrow{p}_i}{|\overrightarrow{p}_i|} \cdot \frac{\Delta\overrightarrow{x}}{|\Delta\overrightarrow{x}|}$, where $\Delta\overrightarrow{x}$ represents displacement of the cell's centroid due to the hypothetical cell-type acceptance described. The polarity updates at each Monte Carlo step (MCS) according to $\Delta\overrightarrow{p}_i = \Delta\overrightarrow{r}_i - \frac{\overrightarrow{p}_i}{t_{memory}}$ where $\Delta\overrightarrow{r}_i$ represents a displacement of the centroid from the previous MCS and $t_{memory}$ sets the decaying rate of the polarity. This update rule represents alignment of the polarity (which can also be viewed as propulsion 'force') to the accumulated moving velocity, in which the constant $1 - \frac{1}{t_{memory}}$ controls the effective number of accumulated velocities: If $t_{memory}\rightarrow\infty$, then the polarity update rule dictates $\overrightarrow{p}_i(t) = \overrightarrow{p}_i(t - 1) + \Delta\overrightarrow{r}_i$ meaning entire history of a cell's trajectory equals the polarity. If $t_{memory} = 1$, which is its minimum value, then the polarity is identical to the instantaneous displacement ($\overrightarrow{p}_i(t) = \Delta\overrightarrow{r}_i$) which is highly stochastic. In the simulation, 1 MCS corresponds to updating sites the number of times equal to the number of total sites in the system.

The set of parameter values we fixed for all CPM simulations are: $t_{memory} = 4$, $\lambda_{area} = 1$, $\lambda_{surface} = 1$, $A_{target} = 100$, $s_{target} = 0.9$, $E_{medium-medium} = 0$, $E_{cell-medium} = 0$, and the temperature was set to 10. Throughout this paper we varied only two parameters: $E = E_{cell-cell}$, the interfacial energy factor representative of the strength of adhesiveness between cells of the same type, and $S$ which controls the strength of the energy associated with the alignment between polarity and velocity as described in [34]. $S$ determines not only the directional persistence but also the migration speed of a cell.

Physical units for the simulation were set as the following. First of all, we set the grid mesh size to be 3 $\mu m$: It was determined based on an average diameter of MDA-MB-231 cells (~ 30 $\mu m$) and the model cell's target area $A_{target}$ which was fixed to 100. Assigning a physical time unit to one MCS before we run CPM simulation was in principle not possible, and it came only after we established the 2 phase diagrams (Fig 7A and 7B) for which we did not need to know the physical time scale beforehand. We identified the set of $S$ and $E$ where two relevant level curves crossed, used the corresponding values of $S$ and $E$ to compute the model cells' instantaneous speed $\overline{v}$, which then was compared with the experimentally measured one, and finally set 1 MSC = 2 min.

## Culture of MDA-MB-231 and sample preparation

Once MDA-MB-231 cells growing on a culture dish in an incubator which maintains 37˚C and 5% of CO2 reached ~ 70% confluency, the cells were trypsinized with Trypsin-EDTA solution 10X (Cat. No. 59418C, Sigma-Aldrich) diluted to 5X with DPBS (Cat. No. D8537, Sigma-Aldrich) for 3 min in the incubator, after removing the DMEM culture media (10% FBS). After being centrifuged, supernatant was discarded, then DPBS was added and well-mixed. This washing procedure was done twice. Subsequently, culture media was added, mixed and counted using hemocytometer. The suspended cells were added on a 35 mm culture-dish and kept in the incubator overnight. For imaging, the sample was prepared by fully adding culture media to the dish and covered it with a thin PDMS to prevent evaporation.

## Time-sequence imaging

Live-cell imaging of the MDA-MB-231 cells was carried out by placing the sample inside a custom-made small, thin cylindrical incubator which was placed on the x-y stage of a microscope (inverted, IX71, Olympus). The incubator was heated by an insulated heating pad which was connected with a temperature controller (SDM9000, Sanup, Korea), while the temperature inside was monitored by a (PT100) thermometer to maintain 37.5 ± 0.1˚C. To prevent water condensation, two indium tin oxide (ITO)-coated windows along the imaging axis were electrically heated. Mixture of CO2 (5%) and air (95%) were continuously perfused to the incubator. Acquisition of the time-sequence images were done using PCO CCD Camera (1600s, Germany) with 4X (NA 0.13) objective lens and Micromanager software. The time-interval between each acquisition was fixed to 2 min throughout the experiments in this study.

## Statistical analysis

Statistical analysis and mathematical fitting was performed using Matlab and home-built softwares. All statistical data were presented as mean ± standard deviation and assessed for statistical significance using a one-way ANOVA.

## Supporting information

**S1 Fig. Mean velocity-velocity correlation along vertical and horizontal axes.** (a) Mean correlation function along the migration (y-) axis (temporal and ensemble mean). Exponential fit is drawn as blue solid line (correlation length, $30.75\mu m$). (b) Correlation function along the axis perpendicular to the migration (x-) axis. The exponential fit had correlation length of $18.08\mu m$.
(TIF)

**S2 Fig. Proliferating MDA-MB-231 cell-population in monolayer showing cluster-forming tendency.** (a) Snapshot images acquired at different time. (b) Ensemble mean of minimum cell-to-cell distance $\langle r_{min} \rangle$ (red) and average inter-particle distance (blue), which is approximated as $\sqrt{A/N}$ where N is the number of cells, and A is the total area. The unit length is $\mu m$. This result illustrates that cells tend to adhere to each other forming small colonies; and this tendency becomes less pronounced as the cell population gets confluent.
(TIF)

**S3 Fig. Probability density function of the number of reverse-turns during 10 hours for rotating cell-doublets (based on n = 57).**
(TIF)

**S4 Fig. Phase diagrams characterizing CPM cell-doublets.** (a) $\overline{d}$: Average distance between pair's centroid and cells. $\overline{d}$ was averaged temporally and over the ensemble of doublets. The white region represents where pairs separate due to strong propulsion overpowering the adhesion strength. (b) Diffusion exponents of angular diffusion (based on time domain: 70 ~ 100 min). For (a) and (b), 200 doublets were analyzed.
(TIF)

**S5 Fig. Phase diagrams characterizing cell motility within a confluent cell-population.** (a) Diffusion exponent $\alpha$ (based on the ensemble mean of MSD, in time domain of 400 ~ 600 min). (b) Normalized (temporal and ensemble) average cluster size $\overline{A}_{cluster}$. Cells belonging to the same cluster were obtained by recursively finding neighbors that satisfy two criteria (first, neighbors should be within 51 $\mu m$ from a reference cell, and second, their velocity vectors should align with that of the reference cell within 20˚).
(TIF)

**S6 Fig. Time-fraction of flocking (co-movement) mode experienced by doublets in the CPM simulations.** Criterion for the flocking mode was set as: 1) mean centroid speed $> 0.45$ $\mu m$/min and 2) mean angular speed $< 0.01 \mu m$/rad. The arrow marks $S = 2.8$.
(TIF)

**S7 Fig. Correlation length along the instantaneous direction of cell movement and mean velocity-velocity correlation maps for cells in confluency.** (a) Correlation length along the migration axis as a function of $S$, which was obtained by fitting the velocity-velocity correlation as in S1B Fig. (b) (Temporal and ensemble) mean velocity-velocity correlation maps for three different $S$. The averaging was done over 400 different reference cells and 60 different times. For each map, the width and height range from -80 $\mu m$ to 80 $\mu m$ with the reference cell at the center. The reference cell's moving direction is aligned along positive y-axis. White-colored pixels represent where there have been no cell visits.
(TIF)

**S8 Fig. MSDs for confluent and freely crawling cells with distributed parameters.** (a) $\overline{S} = 2.8$, $\sigma_s = 1$ with $E = -65$ fixed. (b) $S = 2.8$ fixed with 5 different types of cells + 1 medium, generating 15+1 different $E$s (uniformly distributed ranging from -55 to -75). Blue: confluent population; red: freely crawling cells.
(TIF)

**S1 Movie. MDA-MB-231 cells and their trajectories migrating in a confluent population.**
(AVI)

**S2 Movie. Freely crawling MDA-MB-231 cells.**
(AVI)

**S3 Movie. MDA-MB-231 cell-doublets.**
(AVI)

**S4 Movie. MDA-MB-231 cell-quadruplets.**
(AVI)

**S5 Movie. Simulated doublet, quadruplet, cell-population with trajectories of CPM (E = 0, S = 0).**
(AVI)

**S6 Movie. Simulated doublet, quadruplet, cell-population with trajectories of CPM (E = 0, S = 6).**
(AVI)

**S7 Movie. Simulated doublet, quadruplet, cell-population with trajectories of CPM (E = -100, S = 5).**
(AVI)

**S8 Movie. Simulated doublet, quadruplet, cell-population with trajectories of CPM (E = -100, S = 10).**
(AVI)

## Acknowledgments

We thank Yongjoo Baek and Sung-Gil Chi for discussions and Ellen Lee and Thamara Liz Gabuardi for critical reading of the manuscript.

## Author Contributions

**Conceptualization:** Kyoung J. Lee.

**Data curation:** Kyoung J. Lee.

**Formal analysis:** Hyun Gyu Lee.

**Funding acquisition:** Kyoung J. Lee.

**Investigation:** Hyun Gyu Lee.

**Methodology:** Hyun Gyu Lee.

**Project administration:** Kyoung J. Lee.

**Software:** Hyun Gyu Lee.

**Supervision:** Kyoung J. Lee.

**Validation:** Kyoung J. Lee.

**Visualization:** Hyun Gyu Lee.

**Writing – original draft:** Hyun Gyu Lee, Kyoung J. Lee.

**Writing – review & editing:** Kyoung J. Lee.

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
