## [Decision Letter · Decision Letter 0]

28 Apr 2021

Dear Dr. Lee,

Thank you very much for submitting your manuscript "Neighbor-enhanced diffusivity in dense, cohesive cell populations" for consideration at PLOS Computational Biology.

As with all papers reviewed by the journal, your manuscript was reviewed by members of the editorial board and by several independent reviewers. In light of the reviews (below this email), we would like to invite the resubmission of a significantly-revised version that takes into account the reviewers' comments.

In particular, the reviewers requested a more detailed explanation of the mechanistic assumptions underlying the simulations performed for this study.

We cannot make any decision about publication until we have seen the revised manuscript and your response to the reviewers' comments. Your revised manuscript is also likely to be sent to reviewers for further evaluation.

Sincerely,

Martin Meier-Schellersheim

Associate Editor

PLOS Computational Biology

Jason Haugh

Deputy Editor

PLOS Computational Biology

Reviewer's Responses to Questions

**Comments to the Authors:**

Reviewer #1: The authors present an interesting study, including experimental and simulation data. However, I have several problems that they need to address:

1) There are some missing relevant refs:

Rotations of cell collectives in confinement:

Doxzen, Kevin, et al. "Guidance of collective cell migration by substrate geometry." Integrative biology 5.8 (2013): 1026-1035.‏

Segerer, Felix J., et al. "Emergence and persistence of collective cell migration on small circular micropatterns." Physical review letters 114.22 (2015): 228102.

For freely rotating cellular clusters:

Malet-Engra, Gema, et al. "Collective cell motility promotes chemotactic prowess and resistance to chemorepulsion." Current Biology 25.2 (2015): 242-250.‏

Copenhagen, Katherine, et al. "Frustration-induced phases in migrating cell clusters." Science advances 4.9 (2018): eaar8483.‏‏

2) Its not clear if the characterization of the dynamics by the MSD exponent alpha is very meaningful in the sense of diffusion-vs-super-diffusion, since even a persistent RW eventually gives rise to normal diffusion at long enough time scales.

3) On page 7 they discuss the cell migration. In addition to the analysis that they present, they should plot the velocity-velocity correlations and extract the correlation length, as done for example in [4].

4) They write that: "neighbors have a tendency of moving in the opposite direction of the reference cell is heading.", so do they move in "lanes" ?

They could try the analysis in terms of "co-moving cell clusters", performed in:

Zaritsky, Assaf, et al. "Propagating waves of directionality and coordination orchestrate collective cell migration." PLoS Comput Biol 10.7 (2014): e1003747.‏

5) Could it be that the observed increase in motility when cells interact is related to large flows induced by confluence in cellular nematics ? especially as their cells are highly elongated.

see:

Duclos, G., et al. "Spontaneous shear flow in confined cellular nematics." Nature physics 14.7 (2018): 728-732.‏

Duclos, Guillaume, et al. "Perfect nematic order in confined monolayers of spindle-shaped cells." Soft matter 10.14 (2014): 2346-2353.

6) On page 8 a "modulation period tau_d" is introduced but not defined or explained. Is it the mean time between changes in the direction of rotation ? ie rotational persistence time ? should be clearly explained.

7) Are the doublets ever breaking up ? if they do, what is their life-time distribution ? ‏

8) Points 6,7 also refer to the quadruplets.

9) On page 11 the cell swapping is described along the "normal direction", but a better term is "nearest neighbors along the cluster rim".

10) The biggest problem in this work is the simulations:

It is not explained why in these simulations do they find the observed behavior.

What is the mechanism in the simulations that gives rise to larger diffusivity in confluency ? and larger persistence length and time for the single cell trajectory ? using the simulation they should be able to pinpoint the mechanism, but I dont see it explained anywhere.

Similarly they do not explain why the rotations arise in their simulations.

11) The discussion of Fig.7 is very descriptive. No mechanism is explained, and the region which fits the experiments should be indicated on the figures.

The simulations do not provide us with any understanding of the mechanisms that give rise to the experimental observations, and how these depend on the physical properties of the model, and why.

This is the main purpose of using theoretical simulations, and is lacking here.

Reviewer #2: The review comment is uploaded as an attachment.

Reviewer #3: The authors present a cell-tracking study of MDA-MB-231 cells densely packed in 2-dimensional layer.

The cell trajectories are studied with well established statistical methods in the field, such as MSD, and time-spatial correlations. They report a neighbour-enhanced diffusivity phenomenon where paired neighbour-cells exhibit a higher diffusivity than those of freely moving single cells. Further, they observe cell-doublet and cell-quadruplet rotation. Interesting, the authors were able to reproduce these dynamic properties with a Potts Model by tuning only two parameters: adhesiveness and self-propulsion. I found the findings very interesting. the fact that two parameters of the CPM can account the phenomenon is very instructive about the processes underlying these dynamic properties. However, there are several issues that need to be addressed prior to its recommendation. I will list my concerns below

Major issues

1.- The present form of the Introduction is no appropriate. For example: the text about biological transitions (2nd and 3rd paragraphs lines 64-88) is not well connected with the present results.

It is enough to mention: " ... biological tissues are a collection of interacting ‘active’ cells in non-equilibrium states, therefore, in principle, cell-population can support numerous different states,

to which an initially sedentary tissue state can switch." and appropriate refs. I suggest that the authors include a discussion that sheds light on the title, for example, what are the differences between NED and CEL that justify a new category?

2.- The term super-diffusivity must be restricted to the observational time-scale. If the authors calculate

MSD with a larger number of MC steps in the simulation, they can see that the Brownian motion is replaced by the ballistic regime It is documented in doi: 10.1103/PhysRevE.101.062408 and 10.1103/PhysRevLett.99.010602

The authors must be more rigorous with these terms all over the text.

3.- In lines 429-432, Even in a monoclonal cell population it is expected and heterogeneous. This variability is due to intrinsic and extrinsic stochastic gene expression, a mesoscopic phenomenon well understood at the single-gene level. The sentence in line 432 is not very precise. Moreover, despite

heterogeneous behaviour on cell culture, the authors use a "homogeneous" in-silico cells. I suggest to authors to use a distributed parameter values in the simulation. i.e. each cells in the CPM have intrinsic

value of S, and the same in other simulation regarding the E parameter. In this manner, one can understood the contribution of adhesiveness and/or self-propulsion to the observed variability.

4.- In lines 442-446, the authors discuss previous finding regarding the role of cell-to-cell adhesion in the phenomenon. There is a previous paper which is in line with the present finding (NED) and the role of cell-to-cell adhesion (Fig.2 and 7 of Physica A 365 (2006) 481-490). The paper analysis the cell locomotion of cell that express MOCA protein a protein that change cell adhesion by regulating the ctivity of Rac1 and N-cadherin. The author must to add to the Discussion and Introduction (see point 1)

Minors

a.- The paper emphasises on the fact NED s counter-intuitive dynamic property, I am not agree.

b.- The sentence: "The study of Mitchel et al. is an excellent example suggesting that the relevant phase space for the dense cell population needs to be at least two-dimensional. " What two dimensions are you referring to?

c.- The sentence: "With phenomena like ‘contact inhibition of locomotion’ and ‘volume exclusion’ "

is wrong, "volume exclusion" is not a phenomenon.

d.- I don't understand the lines 46-47.

e.- I feel that the intermittent position change events in quadruplets are collateral,

they do not represent an intrinsic phenomenon, and something is oversized in the text.

Why the authors use the self-propulsion proposed by Szabo et al. instead of the one proposed in 10.3389/fphy.2018.00061?

**Have the authors made all data and (if applicable) computational code underlying the findings in their manuscript fully available?**

Reviewer #1: Yes

Reviewer #2: **No: **

Reviewer #3: Yes

PLOS authors have the option to publish the peer review history of their article (what does this mean?). If published, this will include your full peer review and any attached files.

Reviewer #1: No

Reviewer #2: No

Reviewer #3: **Yes: **Luis Diambra
---

## [Decision Letter · Decision Letter 1]

29 Jun 2021

Dear Dr. Lee,

Thank you very much for submitting your manuscript "Neighbor-enhanced diffusivity in dense, cohesive cell populations" for consideration at PLOS Computational Biology.

As with all papers reviewed by the journal, your manuscript was reviewed by members of the editorial board and by several independent reviewers. In light of the reviews (below this email), we would like to invite the resubmission of a significantly-revised version that takes into account the reviewers' comments.

In particular, there are still significant gaps in the description of the computationally modeled mechanisms leading to the observed cell behavior. Another important aspect that requires clarification is how the current work compares to previously published studies.

We cannot make any decision about publication until we have seen the revised manuscript and your response to the reviewers' comments. Your revised manuscript is also likely to be sent to reviewers for further evaluation.

Sincerely,

Martin Meier-Schellersheim

Associate Editor

PLOS Computational Biology

Jason Haugh

Deputy Editor

PLOS Computational Biology

Reviewer's Responses to Questions

**Comments to the Authors:**

Reviewer #1: The authors have addressed the comments, but some are still in need for clarification, as I list below.

1) The caption for Fig.7 should be expanded, with each panel explained, what are the lines, what is the colorbar indicating etc.

2) The computational model does not include any effect of contact-inhibition of locomotion (CIL) or any tendency for cells to align their polarization with neighbors. Both effects where demonstrated in many multi-cellular systems, so they need to explain and emphasize this fact, and why in these cells they think these interactions are not present.

3) The model shows that where two adhered cells maintain contact, but this contact does not affect their polarization, they rotate. This is quite trivial, considering that two self-propelled particles, even described as point-particles (ie without the CPM detailed description of the cell shapes), can not do anything else under such conditions. The strong binding into a pair, which has to rotate unless the forces are aligned, has therefore a similar effect to the external confinement shown in refs.44,45. They should discuss these issues at greater depth.

4) The CPM model that is used contains a self-propelled polarization vector that is not emergent from the CPM-calculated cell shape, but is external to it. There is not dependence of the polarization on the cell shape. This makes this simulation rather simple-minded. See for example a variant of the CPM where polarization and persistence emerge from the underling dynamics, and therefore motility and persistence are highly shape dependent, which should be mentioned in the discussion of the model that is used and its limitations:

Niculescu, Ioana, Johannes Textor, and Rob J. De Boer. "Crawling and gliding: a computational model for shape-driven cell migration." PLoS Comput Biol 11.10 (2015): e1004280.‏

Wortel, Inge MN, et al. "Local actin dynamics couple speed and persistence in a Cellular Potts Model of cell migration." Biophysical Journal (2021).‏

5) If I understand the model, along the pixels where where cells are adhered to each other, both cells are less likely to make self-propelled moves forward. So these regions are essentially less motile, compared to the outer edges along the outer side of the doublet. If this is true, you have a graded motility that resembles the rim-bulk graded motility that was shown in [37] to give rise to rotating clusters. This can be explained.

In general, I think the authors added references to previous works, but did not attempt to closely compare their model to those previous works. I think they should make an effort to make these comparisons, it will benefit the readers.

Reviewer #2: Thank you very much for answering my questions. The original submitted manuscript has no description on the mathematical model at all, and thus I could not judge the study as mentioned in my previous comments. Now the model description was provided. I found that the model is the same as one previously proposed and this study does not include any methodological novelty. The results are quite descriptive and are not enough to convince their claim. In fact, most of the results in the manuscript (Figure 2,3,5,6) are about cell swirlings that are less directly relevant to the individual cell diffusivity. As I commented before, quantitative description of positional swapping events in the cell clusters in experiments is interesting to some extent. However, this is not the main point for acceptance of this journal. Figure 7 shows key results for this study but it just shows parameter dependencies of produced cellular behaviors in their simplified setting. They changed the parameter value corresponding to the interfacial energy of cell-to-cell, and concluded that the cell-cell adhesion enhances the super-diffusivity of cell migrations only by means of simulations. This is caused by an enhancement of degree of cell spreading in the model, which is not solely provided by the cell-cell adhesion property in reality. The conclusion is intuitive and does not provide profound new biological insights or scientific novelty.

Overall, I do not think this study meets the criteria of this journal.

Reviewer #3: The authors' responses satisfy all my concerns, I have no further comments or criticisms for this manuscript.

**Have the authors made all data and (if applicable) computational code underlying the findings in their manuscript fully available?**

Reviewer #1: **No: **I could not access the SI

Reviewer #2: **No: **

Reviewer #3: **No: **the computational code of the CPM modeling is not available

PLOS authors have the option to publish the peer review history of their article (what does this mean?). If published, this will include your full peer review and any attached files.

Reviewer #1: No

Reviewer #2: No

Reviewer #3: No
---

## [Decision Letter · Decision Letter 2]

13 Sep 2021

Dear Dr. Lee,

We are pleased to inform you that your manuscript 'Neighbor-enhanced diffusivity in dense, cohesive cell populations' has been provisionally accepted for publication in PLOS Computational Biology.

Best regards,

Martin Meier-Schellersheim

Associate Editor

PLOS Computational Biology

Jason Haugh

Deputy Editor

PLOS Computational Biology

Reviewer's Responses to Questions

**Comments to the Authors:**

Reviewer #1: The authors have fully addressed my comments.

Reviewer #2: I understand the authors' claim. Still, I do think that the most of the results are due to the framework of cellular Potts model and may be changed if they use another modeling framework. I have no additional comments for their manuscript.

**Have the authors made all data and (if applicable) computational code underlying the findings in their manuscript fully available?**

Reviewer #1: Yes

Reviewer #2: Yes

PLOS authors have the option to publish the peer review history of their article (what does this mean?). If published, this will include your full peer review and any attached files.

Reviewer #1: No

Reviewer #2: No

---

## [Editor Report · Acceptance letter]

17 Sep 2021

PCOMPBIOL-D-21-00491R2 

Neighbor-enhanced diffusivity in dense, cohesive cell populations

Dear Dr Lee,

I am pleased to inform you that your manuscript has been formally accepted for publication in PLOS Computational Biology. Your manuscript is now with our production department and you will be notified of the publication date in due course.

With kind regards,

Olena Szabo
